# Transition Path Sampling with Improved Off-Policy Training of Diffusion Path Samplers

**Kiyoung Seong**[†], **Seonghyun Park**[†], **Seonghwan Kim, Woo Youn Kim, Sungsoo Ahn**[†]
KAIST
{kyseong98, hyun26, dmdtka00, wooyoun, sungsoo.ahn}@kaist.ac.kr

## Abstract

Understanding transition pathways between two meta-stable states of a molecular system is crucial to advance drug discovery and material design. However, unbiased molecular dynamics (MD) simulations are computationally infeasible because of the high energy barriers that separate these states. Although recent machine learning techniques are proposed to sample rare events, they are often limited to simple systems and rely on collective variables (CVs) derived from costly domain expertise. In this paper, we introduce a novel approach that trains diffusion path samplers (DPS) to address the transition path sampling (TPS) problem without requiring CVs. We reformulate the problem as an amortized sampling from the transition path distribution by minimizing the log-variance divergence between the path distribution induced by DPS and the transition path distribution. Based on the log-variance divergence, we propose learnable control variates to reduce the variance of gradient estimators and the off-policy training objective with replay buffers and simulated annealing techniques to improve sample efficiency and diversity. We also propose a scale-based equivariant parameterization of the bias forces to ensure scalability for large systems. We extensively evaluate our approach, termed TPS-DPS, on a synthetic system, small peptide, and challenging fast-folding proteins, demonstrating that it produces more realistic and diverse transition pathways than existing baselines. We provide links to our project page and code.

## 1 Introduction

In drug discovery and material design, it is crucial to understand the mechanisms of transitions between meta-stable states of molecular systems, such as protein folding (Salsbury Jr, 2010; Piana et al., 2012), chemical reaction (Mulholland, 2005; Ahn et al., 2019), and nucleation (Thanh et al., 2014; Beaupere et al., 2018). Molecular dynamics (MD) simulations have become one of the most widely used tools for sampling these transitions. However, sampling transition paths with unbiased MD simulations is computationally expensive due to high energy barriers, which cause an exponential decay in the probability of making a transition (Pechukas, 1981).

To address this problem, researchers have developed enhanced sampling approaches such as steered MD (SMD; Schlitter et al., 1994; Izrailev et al., 1999), umbrella sampling (Torrie & Valleau, 1977; Kästner, 2011), meta-dynamics (Ensing et al., 2006; Branduardi et al., 2012; Bussi & Branduardi, 2015), adaptive biasing force (ABF; Comer et al., 2015), on-the-fly probability-enhanced sampling (OPES; Invernizzi & Parrinello, 2020) methods. These methods utilize *bias forces* to facilitate transitions across high energy barriers. They require collective variables (CVs), functions of atomic coordinates designed to capture the slow degree of freedom. Although effective for simple systems, the reliance on expensive domain knowledge limits the applicability of the methods to complex systems where CVs are less understood.

Recently, machine learning has emerged as a promising paradigm for CV-free transition path sampling (TPS) (Das et al., 2021; Lelièvre et al., 2023; Holdijk et al., 2024). The key idea is to parameterize the bias force using a neural network and train it to sample transition paths directly via the biased MD simulation. In particular, Lelièvre et al. (2023) considered reinforcement learning to sample paths

---

[†] Work done in POSTECH

escaping meta-stable states. Das et al. (2021); Hua et al. (2024); Holdijk et al. (2024) considered TPS problem as minimizing the reverse Kullback-Leibler (KL) divergence between the path measures induced by the biased MD and the target path measure. However, minimizing the reverse KL divergence suffers from mode collapse, capturing only a subset of modes of the target distribution (Vargas et al., 2023; Richter & Berner, 2024). Furthermore, Das et al. (2021); Lelièvre et al. (2023); Hua et al. (2024); Holdijk et al. (2024) limited their evaluation to low-dimensional synthetic systems or small peptides. Developing machine learning algorithms that generate accurate and diverse transition pathways for complex molecular systems remains an open challenge.

**Contribution.** In this work, we propose the *diffusion path sampler* (DPS) to solve the transition path sampling problem.[1] Our approach, coined TPS-DPS, (1) trains the bias force by minimizing a recently proposed log-variance divergence (Nüsken & Richter, 2021) between the path measure induced by the biased MD and the target path measure, and (2) uses scale-based parameterization of the bias force to handle large systems including fast-folding proteins. Specifically, to leverage desirable properties of the log-variance divergence, such as robustness of gradient estimator and degree of freedom in reference path measure, we propose to learn a control variate for reducing the variance of gradient estimators and employ *off-policy* training scheme with replay buffer and simulated annealing to improve sample efficiency and prevent the mode collapse.

We also introduce a new scale-based equivariant parameterization for the bias force to frequently sample meaningful paths in training. Our key idea is to predict the atom-wise positive scaling factor of displacement from current molecular states to the target meta-stable state. This guarantees the bias force to decrease the distance between them for every MD step. We also use the Kabsch algorithm (Kabsch, 1976) to align the current molecular states with the target meta-stable state, guaranteeing $SE(3)$ equivariance of bias force for better generalization across the states.

We extensively evaluate our method on the synthetic double-well potential with dual channels, Alanine Dipeptide, and four fast-folding proteins: Chignolin, Trp-cage, BBA, and BBL (Lindorff-Larsen et al., 2011). We compare TPS-DPS with the ML approach (PIPS; Holdijk et al., 2024), as well as classical non-ML methods, e.g., steered MD (SMD; Schlitter et al., 1994; Izrailev et al., 1999). Our experiments demonstrate that TPS-DPS consistently generates realistic and diverse transition paths, similar to the ground truth ensemble. In addition, we do ablation studies of the proposed components.

## 2 RELATED WORK

**Transition path sampling without ML.** Metadynamics (Branduardi et al., 2012), on-the-fly probability-enhanced sampling (OPES; Invernizzi & Parrinello, 2020), adaptive biasing force (ABF; Comer et al., 2015), and steered molecular dynamics (SMD; Schlitter et al., 1994; Izrailev et al., 1999) were introduced to explore molecular conformations that are difficult to access by unbiased molecular dynamics (MD) within limited simulation times (Hénin et al., 2022). However, they mostly rely on collective variables (CVs) for high-dimensional problems and are inapplicable to systems with unknown CVs. To sample transition paths without CVs, Dellago et al. (1998) proposed shooting methods that use the Markov chain Monte Carlo (MCMC) procedure on path space. However, these methods often suffer from long mixing times in path space, which hinders the exploration of diverse transition paths. Additionally, high rejection rates can further reduce their efficiency.

**Data driven ML approaches.** Recently, generative models have been used to sample new transition paths given a dataset of transition paths. Petersen et al. (2023); Triplett & Lu (2023) and Lelièvre et al. (2023) applied diffusion probabilistic models (Ho et al., 2020) and variational auto-encoders (Kingma & Welling, 2013) for transition path sampling, respectively. However, these methods are limited to small systems and struggle to collect data using unbiased MD simulations due to the high energy barriers. Klein et al. (2024); Schreiner et al. (2024); Jing et al. (2024) proposed to accelerate MD using time-coarsened dynamics, but the time-coarsened dynamics cannot capture the details of the transition, e.g., the transition states. Duan et al. (2023); Kim et al. (2024) used neural networks to generate transition states of a given chemical reaction, but cannot generate transition paths.

**Data free ML approaches.** Without a previously collected dataset, Das et al. (2021); Lelièvre et al. (2023); Sipka et al. (2023); Hua et al. (2024); Holdijk et al. (2024) trained the bias forces to

---

[1]We coin our method diffusion path sampler since it samples paths using diffusion SDE, similar to diffusion samplers (Zhang & Chen, 2022; Vargas et al., 2023) that use diffusion SDEs for sampling the final state.

sample transition paths using the biased MD. Lelièvre et al. (2023) used reinforcement learning to train the bias forces but focused on escaping an initial meta-stable state rather than targeting a given meta-stable state. Sipka et al. (2023) used differentiable biased MD simulation to train bias potential and introduce partial back-propagation and graph mini-batching techniques to resolve computational issues in differentiable simulation. Das et al. (2021); Hua et al. (2024); Holdijk et al. (2024) considered the TPS problem as minimizing the reverse KL divergence between path distribution from biased MD and transition path distribution. Das et al. (2021); Hua et al. (2024) limited their evaluation to low-dimensional synthetic systems. In this work, we mainly compare our method with (PIPS; Holdijk et al., 2024). Concurrent to our work, Du et al. (2024) considered the TPS problem as minimizing Doob's Lagrangian objective with boundary constraints. They parameterized marginal distribution as (mixture) Gaussian path distribution to satisfy the boundary constraints without simulation in training time and sampled transition paths with the bias force derived from the Fokker-Planck equation in inference time.

## 3 TRANSITION PATH SAMPLING WITH DIFFUSION PATH SAMPLERS

In this section, we introduce our method, coined transition path sampling with diffusion path sampler (TPS-DPS). Our main idea is to formulate the transition path sampling (TPS) problem as a minimization of the log-variance divergence (Nüsken & Richter, 2021) between two path measures: the path measure induced by DPS and that of transition paths. Our main methodological contribution is twofold: (1) a new off-policy training algorithm that minimizes the log-variance divergence with the learnable control variate, replay buffer, and simulated annealing (2) a $SE(3)$ equivariant scale-based parameterization of the bias force that has an inductive bias for dense training signals in large systems.

### 3.1 PROBLEM SETUP

Our goal is to sample transition paths from one meta-stable state to another meta-stable state given a molecular system. We provide an example of the problem for Alanine Dipeptide in Figure 1. We view this as a task to sample paths from an unbiased molecular dynamics (MD) in Equation (1) conditioned on its starting and ending points of initial and target meta-stable states, respectively. To solve this task, we train the bias force parameterized by a neural network to amortize the sampling procedure.

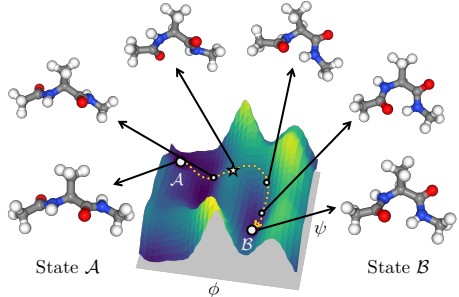

Figure 1: **Problem setup.** The sampled transition path (yellow dotted lines) from the state $\mathcal{A}$ to the state $\mathcal{B}$ on the free energy landscape of Alanine Dipeptide. We visualize the snapshots (white circles) of the transition path and the transition state (white star).

**Molecular dynamics.** We consider a MD simulation on time interval $[0, T]$, i.e., the motion of a molecular state $\boldsymbol{X}_t = (\boldsymbol{R}_t, \boldsymbol{V}_t) \in \mathbb{R}^{6N}$ at time $t$ where $N$ is the number of atoms, $\boldsymbol{R}_t \in \mathbb{R}^{3N}$ is the atom-wise positions and $\boldsymbol{V}_t \in \mathbb{R}^{3N}$ is the atom-wise velocities. In particular, we adopt Langevin dynamics (Bussi & Parrinello, 2007) defined as the following SDE:

$$\mathrm{d}\boldsymbol{X}_t = \boldsymbol{u}(\boldsymbol{X}_t)\mathrm{d}t + \Sigma \mathrm{d}\boldsymbol{W}_t, \quad \boldsymbol{u}(\boldsymbol{X}_t) = \left(\boldsymbol{V}_t, -\frac{\nabla U(\boldsymbol{R}_t)}{\boldsymbol{m}} - \gamma\boldsymbol{V}_t\right), \quad \Sigma = \mathrm{diag}\left(\boldsymbol{\zeta}, \sqrt{\frac{2\gamma k_B \lambda}{\boldsymbol{m}}}\right) \quad (1)$$

where $U, \boldsymbol{m}, \gamma, k_B, \lambda$, and $\boldsymbol{W}_t$ denote the potential energy function, the atom-wise masses, the friction term, the Boltzmann constant, the absolute temperature, and the Brownian motion, respectively, and $\boldsymbol{\zeta} \in \mathbb{R}^{3N}$ is a vector of positive infinitesimal values. MD in Equation (1) induces the path measure, denoted by $\mathbb{P}_0$, which refers to the positive measure defined on measurable subsets of the path space $\mathcal{C}([0, T]; \mathbb{R}^{6N})$ consisting of continuous functions $\boldsymbol{X} : [0, T] \to \mathbb{R}^{6N}$. The path (probability) measure $\mathbb{P}_0$ induced by MD assigns high probability to a set of the probable paths when solving MD.

**Transition path sampling.** One of the challenges in sampling transition paths through unbiased MD simulations is the meta-stability: a state remains trapped for a long time in the initial meta-stable state $\mathcal{A} \subseteq \mathbb{R}^{3N}$ before transitioning into a distinct meta-stable state $\mathcal{B} \subseteq \mathbb{R}^{3N}$. To capture the rare event where transition from $\mathcal{A}$ to $\mathcal{B}$ occurs, we constrain paths $\boldsymbol{X} = (\boldsymbol{X}_t)_{0 \leq t \leq T}$ sampled from

unbiased MD to satisfy $\boldsymbol{R}_0 \in \mathcal{A}$, $\boldsymbol{R}_T \in \mathcal{B}$ for a fixed time $T$. Since the meta-stable state $\mathcal{A}$ and $\mathcal{B}$ are not well-specified for many molecular systems, we simplify this task by (1) fixing a local minima $\boldsymbol{R}_{\mathcal{A}}$, $\boldsymbol{R}_{\mathcal{B}}$ of the potential energy function in the meta-stable states $\mathcal{A}, \mathcal{B}$ and (2) sampling a transition path $\boldsymbol{X}$ that starts from the state $\boldsymbol{R}_0 = \boldsymbol{R}_{\mathcal{A}}$ and ends at the vicinity of $\boldsymbol{R}_{\mathcal{B}}$.

To be specific, we aim to sample from the target path measure $\mathbb{Q}$, which is obtained by reweighting the path measure $\mathbb{P}_0$ with the (normalized) indicator function. The indicator function assigns zero weight to paths that do not reach the vicinity of the target position $\boldsymbol{R}_{\mathcal{B}}$. Formally, the reweighting function is called the Radon–Nikodym derivative defined as follows:

$$\frac{\mathrm{d}\mathbb{Q}}{\mathrm{d}\mathbb{P}_0}(\boldsymbol{X}) = \frac{1_{\mathcal{B}}(\boldsymbol{X})}{Z}, \quad 1_{\mathcal{B}}(\boldsymbol{X}) = \begin{cases} 1 & \text{if } \|\rho_T \cdot \boldsymbol{R}_{\mathcal{B}} - \boldsymbol{R}_T\| \leq \delta, \\ 0 & \text{otherwise,} \end{cases} \quad Z = \mathbb{E}_{\mathbb{P}_0}\left[1_{\mathcal{B}}(\boldsymbol{X})\right], \quad (2)$$

where $\cdot$ denotes group action associated with the $SE(3)$ space and $\rho_T \cdot \boldsymbol{R}_{\mathcal{B}}$ is the aligned target position by the optimal roto-translation $\rho_T \in SE(3)$ to minimize its Euclidean distance to $\boldsymbol{R}_T$, i.e., $\rho_T = \mathrm{argmin}_{\rho \in SE(3)}\|\boldsymbol{R}_T - \rho \cdot \boldsymbol{R}_{\mathcal{B}}\|$. Such a transformation can be obtained from the Kabsch algorithm in $O(N)$ complexity (Kabsch, 1976).

Note that one may consider naïve rejection sampling to sample transition paths, based on running unbiased MD to sample a path $\boldsymbol{X}$ from the path measure $\mathbb{P}_0$ and accepting if the path $\boldsymbol{X}$ arrives at the neighborhood of the position $\boldsymbol{R}_{\mathcal{B}}$ with the radius $\delta$. However, this method struggles to sample transition paths of systems with high energy barriers since the sampled path by unbiased MD rarely reaches the target states, i.e., the rejection ratio is too high.

### 3.2 LOG-VARIANCE MINIMIZATION

In this section, we propose our algorithm to amortize transition path sampling. Our key idea is to train a neural network to induce a path measure that matches the target path measure $\mathbb{Q}$, using the log-variance divergence between the path measures. We propose a new training scheme to minimize the log-variance divergence based on learning the control variate of its gradient and a replay buffer to improve sample efficiency and diversity.

**Amortizing transition path sampling with log-variance divergence.** To match the target path measure $\mathbb{Q}$, we consider a biased MD defined by a policy $\boldsymbol{v}$ (or bias force $\boldsymbol{b}$) as the following SDE:

$$\mathrm{d}\boldsymbol{X}_t = (\boldsymbol{u}(\boldsymbol{X}_t) + \Sigma\boldsymbol{v}(\boldsymbol{X}_t))\mathrm{d}t + \Sigma\mathrm{d}\boldsymbol{W}_t, \quad \boldsymbol{v}(\boldsymbol{X}_t) = \Sigma^{-1}\left(\boldsymbol{0}, \frac{\boldsymbol{b}(\boldsymbol{X}_t)}{\boldsymbol{m}}\right). \quad (3)$$

We also let $\mathbb{P}_{\boldsymbol{v}}$ denote the path measure induced by the SDE. To amortize transition path sampling, we match the path measure $\mathbb{P}_{\boldsymbol{v}_\theta}$ of a parameterized policy $\boldsymbol{v}_\theta$ with the target path measure $\mathbb{Q}$ by minimizing the log-variance divergence:

$$D_{\mathrm{LV}}^{\mathbb{P}}(\mathbb{P}_{\boldsymbol{v}_\theta}\|\mathbb{Q}) = \mathbb{V}_{\mathbb{P}}\left[\log\frac{\mathrm{d}\mathbb{Q}}{\mathrm{d}\mathbb{P}_{\boldsymbol{v}_\theta}}\right] = \mathbb{E}_{\mathbb{P}}\left[\left(\log\frac{\mathrm{d}\mathbb{Q}}{\mathrm{d}\mathbb{P}_{\boldsymbol{v}_\theta}} - \mathbb{E}_{\mathbb{P}}\left[\log\frac{\mathrm{d}\mathbb{Q}}{\mathrm{d}\mathbb{P}_{\boldsymbol{v}_\theta}}\right]\right)^2\right], \quad (4)$$

where $\mathbb{P}$ is an arbitrary reference path measure with $\mathbb{E}_{\mathbb{P}}[\log(\mathrm{d}\mathbb{Q}/\mathrm{d}\mathbb{P}_{\boldsymbol{v}_\theta})] < \infty$. To express the log-variance divergence in detail, we let $\mathbb{P} = \mathbb{P}_{\tilde{\boldsymbol{v}}}$ for some policy $\tilde{\boldsymbol{v}}$ and apply the Girsanov's theorem to Equation (4), deriving the following formulation:

$$D_{\mathrm{LV}}^{\mathbb{P}_{\tilde{\boldsymbol{v}}}}(\mathbb{P}_{\boldsymbol{v}_\theta}\|\mathbb{Q}) = \mathbb{E}_{\mathbb{P}_{\tilde{\boldsymbol{v}}}}\left[(F_{\boldsymbol{v}_\theta, \tilde{\boldsymbol{v}}} - \mathbb{E}_{\mathbb{P}_{\tilde{\boldsymbol{v}}}}[F_{\boldsymbol{v}_\theta, \tilde{\boldsymbol{v}}}])^2\right], \quad (5)$$

$$F_{\boldsymbol{v}_\theta, \tilde{\boldsymbol{v}}}(\boldsymbol{X}) = \frac{1}{2}\int_0^T \|\boldsymbol{v}_\theta(\boldsymbol{X}_t)\|^2\mathrm{d}t - \int_0^T (\boldsymbol{v}_\theta \cdot \tilde{\boldsymbol{v}})(\boldsymbol{X}_t)\mathrm{d}t - \int_0^T \boldsymbol{v}_\theta(\boldsymbol{X}_t) \cdot \mathrm{d}\boldsymbol{W}_t + \log 1_{\mathcal{B}}(\boldsymbol{X}). \quad (6)$$

The first three terms in Equation (6) correspond to the deviation of the biased MD from the unbiased MD integrated over the path sampled from $\mathbb{P}_{\tilde{\boldsymbol{v}}}$. The last term reweights the unbiased MD to the target path measure $\mathbb{Q}$. As a result, minimizing Equation (5) could be thought as minimizing the variation between $\mathbb{P}_{\boldsymbol{v}_\theta}$ and $\mathbb{Q}$. We provide the full derivation in Appendix A.1. Compared to KL divergence, the log-variance divergence provides a robust gradient estimator and avoids differentiating through the SDE solver (Richter et al., 2020; Nüsken & Richter, 2021).

**Minimizing with learnable control variate.** To minimize the log-variance divergence, we consider the following loss that replaces the estimation of $\mathbb{E}_{\mathbb{P}_{\boldsymbol{v}_\theta}}[F_{\boldsymbol{v}_\theta, \boldsymbol{v}_\theta}]$ by learning a scalar parameter $w$:

$$\mathcal{L}(\theta, w) = \mathbb{E}_{\mathbb{P}_{\boldsymbol{v}_\theta}}\left[(F_{\boldsymbol{v}_\theta, \boldsymbol{v}_\theta} - w)^2\right], \quad (7)$$

---

**Algorithm 1** Training

1: Initialize an empty replay buffer $\hat{\mathcal{D}}$, an policy $\boldsymbol{v}_\theta$, a scalar parameter $w$, the number of rollout $I$ and training per rollout $J$, and an annealing schedule $\lambda_{\text{start}} = \lambda_1 > \cdots > \lambda_I = \lambda_{\text{end}}$.
2: **for** $i = 1, \ldots, I$ **do**
3:      Generate $M$ paths $\{\boldsymbol{x}_{0:L}^{(m)}\}_{m=1}^M$ from the biased MD simulations with $\boldsymbol{v}_\theta$ at temperature $\lambda_i$.
4:      Update the replay buffer $\hat{\mathcal{D}} \leftarrow \hat{\mathcal{D}} \cup \{\boldsymbol{x}_{0:L}^{(m)}\}_{m=1}^M$.
5:      **for** $j = 1, \ldots, J$ **do**
6:          Sample $K$ data $\{\boldsymbol{x}_{0:L}^{(k)}\}_{k=1}^K$ from $\hat{\mathcal{D}}$.
7:          Update $\theta$ and $w$ with the gradient of $\frac{1}{K} \sum_{k=1}^K \left( \log \frac{p_0(\boldsymbol{x}_{0:L}^{(k)}) 1_\mathcal{B}(\boldsymbol{x}_{0:L}^{(k)})}{p_{\boldsymbol{v}_\theta}(\boldsymbol{x}_{0:L}^{(k)})} - w \right)^2$.
8:      **end for**
9: **end for**

---

where $w$ is a *control variate* that controls the variance of the gradient estimator of $\nabla_\theta \mathcal{L}(\theta, w)$ without changing the gradient. Note that we set $\tilde{\boldsymbol{v}} = \boldsymbol{v}_\theta$ in Equation (6), which implies that the gradient of Equation (7) coincides with the KL divergence (Richter et al., 2020; Nüsken & Richter, 2021). When optimized, the control variate $w$ estimates the expectation $\mathbb{E}_{\mathbb{P}_{\boldsymbol{v}_\theta}}[F_{\boldsymbol{v}_\theta, \boldsymbol{v}_\theta}]$ since $\text{argmin}_w \mathcal{L}(\theta, w) = \mathbb{E}_{\mathbb{P}_{\boldsymbol{v}_\theta}}[F_{\boldsymbol{v}_\theta, \boldsymbol{v}_\theta}]$. Thus, jointly optimizing $(\theta, w)$ with the gradient step can be interpreted as jointly minimizing log-variance divergence and estimating $\mathbb{E}_{\mathbb{P}_{\boldsymbol{v}_\theta}}[F_{\boldsymbol{v}_\theta, \boldsymbol{v}_\theta}]$ utilizing $w$.

**Off-policy training with replay buffer and simulated annealing.** To leverage the degree of freedom in reference path measure for the log-variance divergence, we allow discrepancy between reference path measure and current path measure, called off-policy training, which is widely used in discrete-time reinforcement learning (Mnih et al., 2013; Bengio et al., 2021). For the sample efficiency, we reuse the samples with a replay buffer $\mathcal{D}$ which stores path samples from the path measure $\mathbb{P}_{\boldsymbol{v}_{\bar{\theta}}}$ associated with previous policies $\boldsymbol{v}_{\bar{\theta}}$. Our modified loss function $\mathcal{L}^\mathcal{D}$ with $\mathcal{D}$ is defined as follows:

$$\mathcal{L}^\mathcal{D}(\theta, w) = \mathbb{E}_{(\boldsymbol{v}_{\bar{\theta}}, \boldsymbol{X}) \sim \mathcal{D}}[(F_{\boldsymbol{v}_\theta, \boldsymbol{v}_{\bar{\theta}}}(\boldsymbol{X}) - w)^2]. \tag{8}$$

The replay buffer also prevents mode collapse, using diverse paths from different path measures. Additionally, in line with other off-policy training algorithms (Malkin et al., 2022; Kim et al., 2023), we utilize the simulated annealing technique to sample diverse paths that cross high-energy barriers.

**Discretization.** To implement the algorithm, we discretize Equation (8). Given a discretization step size $\Delta t$, we consider the discretized paths $\boldsymbol{x}_{0:L} = (\boldsymbol{x}_0, \boldsymbol{x}_1, \ldots, \boldsymbol{x}_L)$ of $\boldsymbol{X}$ from MD simulations where $L = T/\Delta t$ and $\boldsymbol{x}_\ell = \boldsymbol{X}(\ell \Delta t)$. In discrete cases, the discretized paths $\boldsymbol{x}_{0:L}$ from previous policies $\boldsymbol{v}_{\bar{\theta}}$ and their (gradient-detached) policy values $(\boldsymbol{v}_{\bar{\theta}}(\boldsymbol{x}_0), ..., \boldsymbol{v}_{\bar{\theta}}(\boldsymbol{x}_L))$ are used to approximate the value $F_{\boldsymbol{v}_\theta, \boldsymbol{v}_{\bar{\theta}}}(\boldsymbol{X})$ in Equation (6) as follows:

$$\hat{F}_{\boldsymbol{v}_\theta, \boldsymbol{v}_{\bar{\theta}}}(\boldsymbol{x}_{0:L}) = \frac{1}{2} \sum_{\ell=0}^{L-1} \|\boldsymbol{v}_\theta(\boldsymbol{x}_\ell)\|^2 \Delta t - \sum_{\ell=0}^{L-1} (\boldsymbol{v}_\theta \cdot \boldsymbol{v}_{\bar{\theta}})(\boldsymbol{x}_\ell) \Delta t - \sum_{\ell=0}^{L-1} \boldsymbol{v}_\theta(\boldsymbol{x}_\ell) \cdot \boldsymbol{\epsilon}_\ell + \log 1_\mathcal{B}(\boldsymbol{x}_{0:L}), \tag{9}$$

where the noise $\boldsymbol{\epsilon}_\ell = \Sigma^{-1}(\boldsymbol{x}_{\ell+1} - \boldsymbol{x}_\ell - (\boldsymbol{u}(\boldsymbol{x}_\ell) + \Sigma \boldsymbol{v}_{\bar{\theta}}(\boldsymbol{x}_\ell))\Delta t)$ is the discretized Brownian motions of the Langevin dynamics with policy $\boldsymbol{v}_{\bar{\theta}}$. For implementation, we further derive a simple discretized loss of Equation (8) from Equation (9) as follows:

$$\mathbb{E}_{\boldsymbol{x}_{0:L} \sim \hat{\mathcal{D}}} \left[ \left( \log \frac{p_0(\boldsymbol{x}_{0:L}) 1_\mathcal{B}(\boldsymbol{x}_{0:L})}{p_{\boldsymbol{v}_\theta}(\boldsymbol{x}_{0:L})} - w \right)^2 \right], \tag{10}$$

where the buffer $\hat{\mathcal{D}}$ stores paths $\boldsymbol{x}_{0:L}$ sampled from the previous policies, and $p_0$ and $p_{\boldsymbol{v}_\theta}$ denote discrete time transition probability induced by Equations (1) and (3), respectively. We provide a formal derivation of the discretized loss in Appendix A.2. Note that the same objective was derived in the name of relative trajectory balance by Venkatraman et al. (2024).

We describe our training algorithm in Algorithm 1. Overall, our off-policy training algorithm iterates through four steps: (1) sampling paths from the biased MD simulation with current policy $\boldsymbol{v}_\theta$ at high temperature, (2) storing sampled paths in the replay buffer $\hat{\mathcal{D}}$, (3) sampling a batch of the paths from the replay buffer, and (4) training current policy $\boldsymbol{v}_\theta$ by minimizing the loss in Equation (10). After training, the biased MD simulation can directly sample transition paths from the target path measure.

### 3.3 PARAMETERIZATION FOR LARGE SYSTEMS

In this section, we introduce new parameterizations of the bias force and the indicator function to sample transition paths of large systems. Our parameterization is designed to alleviate the problem of sparse training signal, where the model struggles to collect meaningful paths that end at the vicinity of the target meta-stable state in training. This problem is especially severe in large systems.

**Bias force parameterization.** To frequently sample the meaningful paths, we aim to parameterize the bias force which guarantees to reduce the distance between the current molecular state and the target meta-stable state for every MD step. This is achieved by predicting the atom-wise positive scaling factor of the direction to the aligned target meta-stable state rather than predicting force or potential directly. Moreover, we design the bias force to satisfy roto-translational equivariance to the current molecular state input $X_t$, aligning with the symmetry of the transition path sampling problem for better generalization and parameter efficiency.

To be specific, we achieve the inductive bias for dense training signals and the $SE(3)$ equivariance to the current molecular state input $X_t$ by parameterizing the bias force as follows:

$$b(X_t) = \text{diag}(s_\theta(\rho_t^{-1} \cdot X_t))(\rho_t \cdot R_\mathcal{B} - R_t), \tag{11}$$

where $s_\theta(\cdot) \in \mathbb{R}_+^{3N}$ is a neural network constrained to have positive output elements and predicts atom-wise scaling factors and the optimal roto-translation $\rho_t = \text{argmin}_{\rho \in SE(3)} \|R_t - \rho \cdot R_\mathcal{B}\|$ which aligns $R_\mathcal{B}$ with $R_t$, as in Equation (2). We note that the bias force (divided by atom-wise masses) is positively correlated with the direction to target state, i.e., $(b(X_t)/m)^\top (\rho_t \cdot R_\mathcal{B} - R_t) > 0$.

To formalize the benefit of the positive correlation between the bias force and the direction to the target state, one can prove that there always exists a small enough step size $\Delta t$ that decreases the distance between the current state $R_t$ and the aligned target state $\rho'_{t+\Delta t} \cdot R_\mathcal{B}$, i.e.,

$$\|\rho'_{t+\Delta t} \cdot R_\mathcal{B} - R'_{t+\Delta t}\| < \|\rho_t \cdot R_\mathcal{B} - R_t\|, \tag{12}$$

where $R'_{t+\Delta t} = R_t + b(X_t)\Delta t/m$ is the position updated by the bias force with step size $\Delta t$ and $\rho'_{t+\Delta t} = \text{argmin}_{\rho \in SE(3)} \|R'_{t+\Delta t} - \rho \cdot R_\mathcal{B}\|$. We provide the proof of Equation (12) in Appendix A.3.

In the experiments, we also consider other equivariant parameterizations that are less constrained: (1) directly predicting the equivariant bias force $b(X_t) = \rho_t \cdot b_\theta(\rho_t^{-1} \cdot X_t) \in \mathbb{R}^{3N}$ and (2) predicting the invariant bias potential $b_\theta(\rho_t^{-1} \cdot X_t) \in \mathbb{R}$ to obtain the bias force $b(X_t) = -\nabla b_\theta(\rho_t^{-1} \cdot X_t) \in \mathbb{R}^{3N}$. We observe these two parameterizations to be useful for low-dimensional tasks but struggle to produce meaningful paths in large systems during training. As shown in Figure 2, bias forces with the positive scaling parameterization are positively correlated with the direction to the target position regardless of network parameters, unlike direct force parameterizations.

**Indicator function parameterization.** We propose to relax the indicator function $1_\mathcal{B}$ into a radial basis function (RBF) kernel $\tilde{1}_\mathcal{B}(X) = k(R_T, \rho^{-1} \cdot R_\mathcal{B}; \sigma^2)$ which measures the similarity between two positions where $\sigma > 0$ controls the degree of relaxation. The range of RBF kernel $k$ is bounded by the interval $(0, 1]$ so that $\log \tilde{1}_\mathcal{B}(X)$ is well-defined and $\tilde{1}_\mathcal{B}(X)$ represents the binary indicator function smoothly. To capture a high training signal from subtrajectories of sampled paths, we propose to take maximum over RBF kernel values of all intermediate states by $\tilde{1}_\mathcal{B}^{\max}(X) = \max_{t \in [0,T]} k(R_t, \tilde{R}_\mathcal{B}; \sigma^2)$. To extract the subtrajectory with a high training signal, we can truncate the paths at the time that maximizes RBF kernel values, allowing variable path lengths. Notably, the relaxed indicator function is $SE(3)$ invariant to $R_t$ because of the Kabsch algorithm.

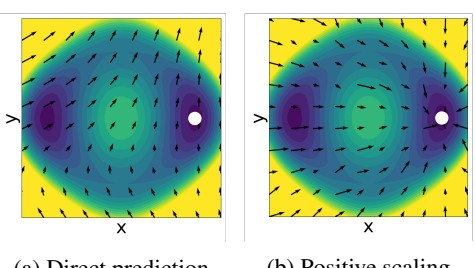

(a) Direct prediction    (b) Positive scaling

Figure 2: **Visualization of the bias force fields of two different bias force parameterizations with initialized neural networks.** **(a)** directly predicting the bias force and **(b)** predicting the positive scaling factors of direction to the target position (white circle).

Table 1: **Benchmark scores on the double-well system and Alanine Dipeptide**. All metrics are averaged over 1024 paths for the double-well system, and 64 paths for Alanine Dipeptide. ETS is computed for paths that hit the target meta-stable state, and the best results are highlighted in **bold**. Predicting the bias force, potential, and atom-wise positive scaling are denoted by (F), (P), and (S), respectively. UMD ($\lambda$) denotes unbiased MD with temperature $\lambda$ and SMD ($k$) denotes steered MD with the force constant $k$. Unless otherwise specified, paths are generated by MD simulation at 1200K for double-well and 300K for Alanine Dipeptide.

| Method | RMSD ($\downarrow$) Å | THP ($\uparrow$) % | ETS ($\downarrow$) kJmol$^{-1}$ | Method | RMSD ($\downarrow$) Å | THP ($\uparrow$) % | ETS ($\downarrow$) kJmol$^{-1}$ |
|---|---|---|---|---|---|---|---|
| Double-well | | | | Alanine Dipeptide | | | |
| UMD (1200K) | $2.21 \pm 0.10$ | 0.00 | - | UMD (300K) | $1.59 \pm 0.15$ | 0.00 | - |
| UMD (2400K) | $2.11 \pm 0.38$ | 3.03 | $1.69 \pm 0.31$ | UMD (3600K) | $1.19 \pm 0.32$ | 6.25 | $812.47 \pm 148.80$ |
| UMD (3600K) | $1.85 \pm 0.68$ | 12.60 | $2.12 \pm 0.41$ | SMD (10) | $0.86 \pm 0.21$ | 7.81 | $33.15 \pm 6.46$ |
| UMD (4800K) | $1.54 \pm 0.81$ | 21.58 | $2.77 \pm 0.69$ | SMD (20) | $0.56 \pm 0.27$ | 54.69 | $78.40 \pm 12.76$ |
| SMD (0.5) | $0.98 \pm 0.90$ | 52.15 | $1.54 \pm 0.21$ | PIPS (F) | $0.66 \pm 0.15$ | 43.75 | $28.17 \pm 10.86$ |
| SMD (1) | $0.14 \pm 0.08$ | 99.80 | $1.85 \pm 0.16$ | PIPS (P) | $1.66 \pm 0.03$ | 0.00 | - |
| TPS-DPS (F, Ours) | $\mathbf{0.01 \pm 0.02}$ | **99.90** | $1.38 \pm 0.16$ | TPS-DPS (F, Ours) | $\mathbf{0.16 \pm 0.06}$ | **92.19** | $19.82 \pm 15.88$ |
| TPS-DPS (P, Ours) | $\mathbf{0.01 \pm 0.03}$ | 99.71 | $\mathbf{1.36 \pm 0.15}$ | TPS-DPS (P, Ours) | $\mathbf{0.16 \pm 0.10}$ | 87.50 | $\mathbf{18.37 \pm 10.86}$ |
| TPS-DPS (S, Ours) | $\mathbf{0.01 \pm 0.03}$ | 99.80 | $1.73 \pm 0.20$ | TPS-DPS (S, Ours) | $0.25 \pm 0.20$ | 76.00 | $22.79 \pm 13.57$ |

$U$ $\qquad$ $b_\theta$ $\qquad$ $U + b_\theta$

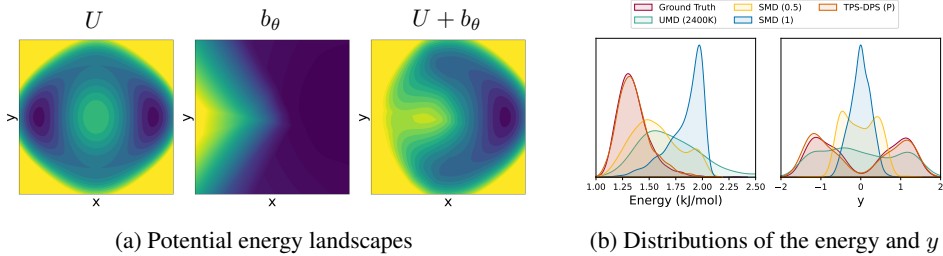

(a) Potential energy landscapes $\qquad\qquad$ (b) Distributions of the energy and $y$

Figure 3: **Visualization of potential energy landscapes and distributions in double-well potential.** (a) Visualization of the learned bias potential $b_\theta$ of TPS-DPS (P). (b) Distributions of the potential energy and $y$ coordinate of transition states from 1024 transition paths sampled by each method.

## 4 EXPERIMENT

In this section, we compare our method, termed TPS-DPS, with both classical non-ML and ML approaches, assessing the accuracy and diversity of sampled transition paths. We begin with the double-well system and Alanine Dipeptide followed by the four fast-folding proteins: Chignolin, Trp-cage, BBA, and BBL. Additionally, we conduct ablation studies to validate the effectiveness of each component in our method. All real-world molecular systems are simulated using the OpenMM library (Eastman et al., 2023). Details on OpenMM simulation and model configurations are provided in Appendices B.1 and B.2, respectively. In Appendix C, we analyze the time complexity of TPS-DPS and evaluate the number of energy evaluations and runtime in training and inference time.

**Evaluation Metrics.** We consider three metrics to evaluate models: (Kabsch) RMSD, THP, and ETS. The root mean square distance (RMSD) measures the ability of the model to produce final positions of paths close to the target position $\boldsymbol{R}_\mathcal{B}$. The target hit percentage (THP) measures the ability of the model to produce final positions of paths that successfully arrive at the target meta-stable state $\mathcal{B}$. Finally, the energy of the transition state (ETS) measures the ability of the model to identify probable transition states. For further details, refer to Appendix B.3.

**Baselines.** We compare TPS-DPS with both non-ML and ML baselines. For non-ML baselines, we consider unbiased MD (UMD) with various temperatures and steered MD (SMD; Schlitter et al., 1994; Izrailev et al., 1999) with various force constants and collective variables (CVs). For ML baselines, we consider a CV-free transition path sampling method, path integral path sampling (PIPS; Holdijk et al., 2024) which also trains a bias force by minimizing the KL divergence between path measures induced by the biased MD and the target path measure.

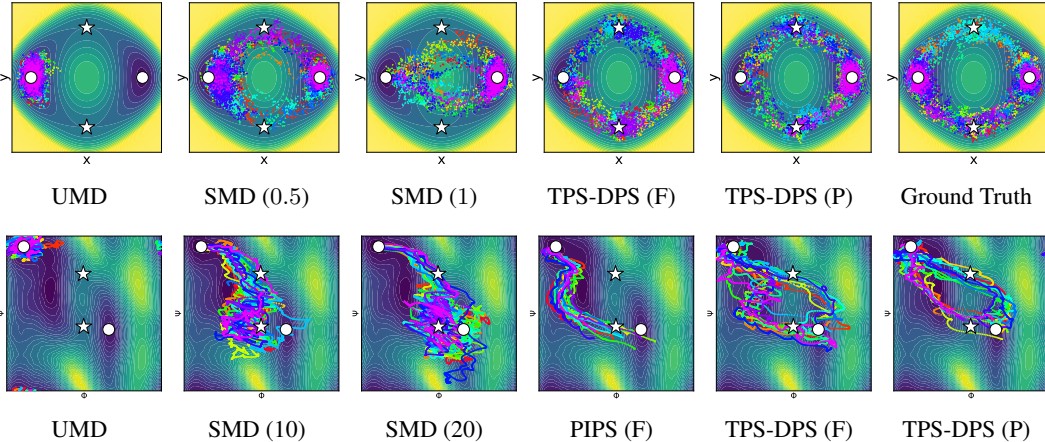

Figure 4: **Visualization of sampled paths on energy landscapes.** For the double-well system, we aim to sample transition paths from the left meta-stable state to the right on the potential energy landscape (top). For Alanine Dipeptide, we aim to sample conformational changes from the $C5$ (upper left) to the $C7ax$ (lower right) on the Ramachandran plot (bottom). White circles and stars indicate meta-stable states and saddle points, respectively.

## 4.1 DOUBLE-WELL SYSTEM

We begin by evaluating our method on a synthetic two-dimensional double-well system with dual channels. This system has two global minima representing the meta-stable states and two reaction pathways with the corresponding saddle points. We aim to sample transition paths from the left state $R_{\mathcal{A}}$ to the right meta-stable states $\mathcal{B} = \{R \mid \|R - R_{\mathcal{B}}\| < 0.5\}$. We collect ground truth path ensembles by rejection sampling which proposes paths sampled from the unbiased MD simulations and accepts if the final states are in the target meta-stable states $\mathcal{B}$. We provide more details on the double-well system in Appendix B.4.

As shown in Table 1, Figure 3, and Figure 4, TPS-DPS outperforms baselines regardless of the bias force parameterizations and generates more similar transition paths to the ground truth than baselines. In Figure 3, the bias potential accelerates the transition by increasing the potential energy near the initial meta-stable state while decreasing the potential energy near the two energy barriers. Moreover, the distribution of energy and $y$ coordinate of the transition states from TPS-DPS is closest to the ground truth compared with other baselines, successfully capturing two reaction channels. In Figure 4, TPS-DPS (F) and (P) generate similar transition paths to the ground truth while UMD at 1200K fails to escape the initial state and SMD struggles to pass the saddle points.

## 4.2 ALANINE DIPEPTIDE

For real-world molecules, we first consider Alanine Dipeptide consisting of two alanine residues and aim to sample conformational changes from the meta-stable state $C5$ to $C7ax$ as shown in Figure 4. The target meta-stable states are defined as $\mathcal{B} = \{R \mid \|\xi(R) - \xi(R_{\mathcal{B}})\| < 0.75\}$, where $\xi(R) = (\phi, \psi)$ is a well-known collective variable which consists of two backbone dihedral angles. Alanine Dipeptide has two reaction channels with the corresponding saddle points.

In Table 1 and Figure 4, our method shows superior performance regardless of the bias force parameterizations and successfully generates diverse transition paths that capture two reaction channels. Compared to our method, UMD at 300K fails to escape the initial state, SMD with the two backbone torsion CV generates transition paths with less probable transition states, and two-way shooting struggles to find plausible transition states. PIPS generates transition paths with only one reaction channel, suffering from mode collapse.

Table 2: **Benchmark scores on fast-folding proteins.** All metrics are averaged over 64 paths. Unless otherwise specified, paths are generated at 300K for Chignolin and 400K for the others.

| Method | RMSD ($\downarrow$) Å | THP ($\uparrow$) % | ETS ($\downarrow$) kJmol$^{-1}$ | Method | RMSD ($\downarrow$) Å | THP ($\uparrow$) % | ETS ($\downarrow$) kJmol$^{-1}$ |
|---|---|---|---|---|---|---|---|
| | Chignolin | | | | Trp-cage | | |
| UMD (300K) | $7.98 \pm 0.41$ | 0.00 | - | UMD (400K) | $7.94 \pm 0.65$ | 0.00 | - |
| UMD (1200K) | $7.23 \pm 0.93$ | 1.56 | 388.17 | UMD (1200K) | $8.27 \pm 1.13$ | 0.00 | - |
| SMD (10k) | $1.26 \pm 0.31$ | 6.25 | $-527.95 \pm 93.58$ | SMD (10k) | $1.68 \pm 0.23$ | 3.12 | $-312.54 \pm 20.67$ |
| SMD (20k) | $\mathbf{0.85 \pm 0.24}$ | 34.38 | $179.52 \pm 138.87$ | SMD (20k) | $1.20 \pm 0.20$ | 42.19 | $-226.40 \pm 85.59$ |
| PIPS (F) | $4.66 \pm 0.17$ | 0.00 | - | PIPS (F) | $7.47 \pm 0.19$ | 0.00 | - |
| PIPS (P) | $4.67 \pm 0.32$ | 0.00 | - | PIPS (P) | $6.07 \pm 0.26$ | 0.00 | - |
| TPS-DPS (F, Ours) | $4.41 \pm 0.49$ | 0.00 | - | TPS-DPS (F, Ours) | $6.35 \pm 0.31$ | 0.00 | - |
| TPS-DPS (P, Ours) | $3.87 \pm 0.42$ | 0.00 | - | TPS-DPS (P, Ours) | $3.15 \pm 0.52$ | 12.50 | $\mathbf{-512.97 \pm 56.89}$ |
| TPS-DPS (S, Ours) | $1.17 \pm 0.66$ | **59.38** | $\mathbf{-780.18 \pm 216.93}$ | TPS-DPS (S, Ours) | $\mathbf{0.76 \pm 0.12}$ | **81.25** | $-317.61 \pm 140.89$ |
| | BBA | | | | BBL | | |
| UMD (400K) | $10.03 \pm 0.39$ | 0.00 | - | UMD (400K) | $18.48 \pm 0.63$ | 0.00 | - |
| UMD (1200K) | $10.81 \pm 1.05$ | 0.00 | - | UMD (1200K) | $18.90 \pm 1.16$ | 0.00 | - |
| SMD (10k) | $2.89 \pm 0.32$ | 0.00 | - | SMD (10k) | $3.67 \pm 0.22$ | 0.00 | - |
| SMD (20k) | $1.66 \pm 0.30$ | 26.56 | $-3104.95 \pm 97.57$ | SMD (20k) | $2.97 \pm 0.33$ | 7.81 | $-1738.57 \pm 386.81$ |
| PIPS (F) | $9.84 \pm 0.18$ | 0.00 | - | PIPS (F) | $17.92 \pm 0.29$ | 0.00 | - |
| PIPS (P) | $9.09 \pm 0.36$ | 0.00 | - | PIPS (P) | $12.67 \pm 0.31$ | 0.00 | - |
| TPS-DPS (F, Ours) | $9.48 \pm 0.18$ | 0.00 | - | TPS-DPS (F, Ours) | $10.15 \pm 0.54$ | 0.00 | - |
| TPS-DPS (P, Ours) | $3.89 \pm 0.35$ | 0.00 | - | TPS-DPS (P, Ours) | $6.45 \pm 0.26$ | 0.00 | - |
| TPS-DPS (S, Ours) | $\mathbf{1.21 \pm 0.09}$ | **84.38** | $\mathbf{-3801.68 \pm 139.38}$ | TPS-DPS (S, Ours) | $\mathbf{1.60 \pm 0.19}$ | **43.75** | $\mathbf{-3616.32 \pm 213.66}$ |

(a) Alanine Dipeptide

(b) Chignolin

(c) Visualization of Chignolin

Figure 5: **Qualitative evaluation on transition path sampled from TPS-DPS. (a)** Potential energy and the two backbone dihedral angle distances between the current and target states. **(b)** Potential energy and the two hydrogen bond distances between the current and the target state. **(c)** Visualization of hydrogen bond formation in Chignolin. We highlight each hydrogen bond in green and yellow.

## 4.3 FAST-FOLDING PROTEINS

Finally, for challenging molecules, we consider four fast-folding proteins (Lindorff-Larsen et al., 2011): Chignolin, Trp-cage, BBA, and BBL with 10, 20, 28, and 47 amino acids, respectively. Proteins fold into stable structures by forming networks of hydrogen bonds. We aim to sample protein folding processes as seen in Figure 7. We define the target meta-stable state as $\mathcal{B} = \{\boldsymbol{R} \mid \|\xi(\boldsymbol{R}) - \xi(\boldsymbol{R}_{\mathcal{B}})\| < 0.75\}$ where $\xi$ consists of the top two components of time-lagged independent component analysis (TICA; Pérez-Hernández et al., 2013). We further describe TICA in Appendix B.4.

As shown in Table 2 and Figure 7, only TPS-DPS (S) successfully samples transition paths that pass probable transition states while TPS-DPS (F) and TPS-DPS (P) fail to hit the target meta-stable state due to the lack of meaningful training signal. While SMD hits the target meta-stable, its transition paths pass less probable transition states. UMD and PIPS fail to hit the target meta-stable state. In Figure 5, we validate the sampled paths using the potential energy and donor-acceptor distance of the two key hydrogen bonds in Chignolin folding, ASP3OD-THR6OG and ASP3N-THR8O used in (Yang et al., 2024). The sampled path of reduces the donor-acceptor distance below the threshold 3.5Å. For 3D videos of transition paths of fast-folding proteins, we refer to project page.

| Method | RMSD (↓) Å | THP (↑) % | ETS (↓) kJmol$^{-1}$ |
|---|---|---|---|
| **Ours** | **0.16 ± 0.06** | **92.19** | **19.82 ± 15.88** |
| w/ KL | 0.43 ± 0.34 | 53.12 | 27.88 ± 14.38 |
| w/ local | 0.24 ± 0.15 | 73.44 | 22.53 ± 14.45 |
| w/o replay | 0.33 ± 0.27 | 64.06 | 24.38 ± 12.31 |
| w/o annealing | 0.67 ± 0.21 | 9.38 | 69.86 ± 30.15 |
| w/o various len | 0.23 ± 0.11 | 75.00 | 29.49 ± 14.13 |
| w/o equivariance | 0.34 ± 0.17 | 56.25 | 22.12 ± 16.96 |

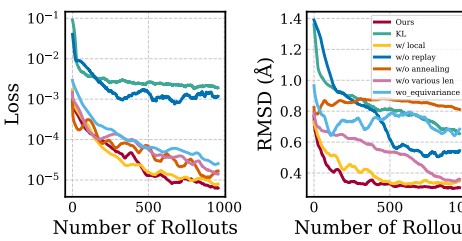

(a) Component-wise performance on Alanine Dipeptide.

(b) Loss and RMSD curves over rollouts.

Figure 6: **Ablation studies on each proposed component of TPS-DPS (F) in Alanine Dipeptide.** **(a)** Benchmark scores on Alanine Dipeptide. **(b)** Loss and RMSD curves averaged over 8 seeds.

## 4.4 ABLATION STUDY

We conduct ablation studies to verify the effectiveness of the six proposed components: log-variance loss, learnable control variate, replay buffer, simulated annealing, maximum over RBF values for various path lengths, and $SE(3)$ equivariance. To be specific, we (1) replace the log-variance divergence with the KL divergence, (2) replace the learnable control variate $w \in \mathbb{R}$ with the local control variate which is Monte-Carlo estimator used in Nüsken & Richter (2021), (3) remove the replay buffer and rely solely on data from the current policy, (4) use only one temperature $\lambda = 300K$ in sampling, (5) remove maximum operation over RBF kernel values using only the final state, (6) skip the alignment of neural network input states with the target position by the Kabsch algorithm.

As seen in Figure 6, all the proposed components improve performance. Our loss is smaller than the KL divergence by more than two orders of magnitude and significantly improves performance. Learning the control variate slightly improves performance, showing that utilizing data from previous policies is effective. The replay buffer significantly improves training efficiency, and shows that the large performance gap between our loss and KL divergence comes from the replay buffer. Simulated annealing for biased MD simulation is critical to finding transition paths. RMSD does not decrease without simulated annealing while loss decreases significantly. For the relaxed indicator function, maximum operation accelerates convergence and improves performance with frequent training signals from the subtrajectories. Leveraging the symmetry of the bias force with the Kabsch algorithm improves performance. We further compare with reverse KL divergence in Appendix E.

## 5 CONCLUSION

In this work, we introduced a novel CV-free diffusion path sampler, called TPS-DPS, to amortize the cost of sampling transition paths. We propose the log-variance divergence with the learnable control variate and off-policy training with the replay buffer and simulated annealing. We propose a new scale-based equivariant parameterization of bias force and relaxed indicator function for reaching target meta-stable states and frequent training signals, particularly in large molecules. Evaluations on double-well, Alanine Dipeptide, and four fast-folding proteins demonstrate that TPS-DPS is superior in accuracy and diversity compared to non-ML and ML approaches.

**Limitation.** While our experiments show promise, they are limited to small fast-folding proteins (up to 50 amino acids). The applicability of our method to real-world proteins with more than 500 amino acids remains unexplored. Additionally, integrating various libraries, such as PLUMED (plu, 2019) and DMFF (Wang et al., 2023), for MD with neural network bias force, has not yet been investigated.

Our method does not generalize across unseen pairs of meta-stable states or different molecular systems. These points to an interesting venue for future research, which would be more appealing for practical applications in drug discovery or material design.

ACKNOWLEDGEMENTS

This work partly was supported by Institute for Information & communications Technology Technology Planning & Evaluation(IITP) grant funded by the Korea government(MSIT)(RS-2019-II190075), Artificial Intelligence Graduate School Support Program(KAIST) and the National Research Foundation of Korea(NRF) grant funded by the Ministry of Science and ICT(MSIT) (No. RS-2022-NR072184), and GRDC(Global Research Development Center) Cooperative Hub Program through the National Research Foundation of Korea(NRF) grant funded by the Ministry of Science and ICT(MSIT) (No. RS-2024-00436165).

We sincerely thank Soojung Yang for providing helpful feedback and suggestions in preparing the early version of the manuscript.

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

# A  METHOD DETAILS

## A.1  LOG VARIANCE FORMULATION

In this section, we derive Equation (5) from Equation (4) to get the explicit expression for log-variance divergence in terms of SDE in Equation (1) and Equation (3). We refer to Nüsken & Richter (2021, Appendix A.1) for the derivation in more general settings.

Our goal is to derive that

$$\mathbb{E}_{\mathbb{P}_{\tilde{v}}}\left[\left(\log\frac{\mathrm{d}\mathbb{Q}}{\mathrm{d}\mathbb{P}_{v_\theta}} - \mathbb{E}_{\mathbb{P}_{\tilde{v}}}\left[\log\frac{\mathrm{d}\mathbb{Q}}{\mathrm{d}\mathbb{P}_{v_\theta}}\right]\right)^2\right] = \mathbb{E}_{\mathbb{P}_{\tilde{v}}}\left[(F_{v_\theta,\tilde{v}} - \mathbb{E}_{\mathbb{P}_{\tilde{v}}}[F_{v_\theta,\tilde{v}}])^2\right], \tag{13}$$

To this end, we focus on calculating $\log\frac{\mathrm{d}\mathbb{Q}}{\mathrm{d}\mathbb{P}_{v_\theta}}(\boldsymbol{X})$ when $\boldsymbol{X} \sim \mathbb{P}_{\tilde{v}}$. Following (Nüsken & Richter, 2021, Lemma A.1), we apply Girsanov's Theorem to calculate the Radon-Nikodym derivative $\frac{\mathrm{d}\mathbb{P}_{v_\theta}}{\mathrm{d}\mathbb{P}_0}$ as follows:

$$\frac{\mathrm{d}\mathbb{P}_{v_\theta}}{\mathrm{d}\mathbb{P}_0}(\boldsymbol{X}) = \exp\left(\int_0^T (\boldsymbol{v}_\theta^T \Sigma^{-1})(\boldsymbol{X}_t) \cdot \mathrm{d}\boldsymbol{X}_t - \int_0^T (\Sigma^{-1}\boldsymbol{u} \cdot \boldsymbol{v}_\theta)(\boldsymbol{X}_t)\mathrm{d}t - \frac{1}{2}\int_0^T \|\boldsymbol{v}_\theta(\boldsymbol{X}_t)\|^2\mathrm{d}t\right). \tag{14}$$

Since the state $\boldsymbol{X}_t$ follows the SDE $\mathrm{d}\boldsymbol{X}_t = (\boldsymbol{u}(\boldsymbol{X}_t) + \Sigma\tilde{\boldsymbol{v}}(\boldsymbol{X}_t))\mathrm{d}t + \Sigma\mathrm{d}\boldsymbol{W}_t$. We plug it into Equation (14) and utilize the definition of the target path measure $\mathbb{Q}$ in Equation (2) to compute $\log\frac{\mathrm{d}\mathbb{Q}}{\mathrm{d}\mathbb{P}_0}$ as follows:

$$\log\frac{\mathrm{d}\mathbb{Q}}{\mathrm{d}\mathbb{P}_{v_\theta}}(\boldsymbol{X}) = \log\frac{\mathrm{d}\mathbb{Q}}{\mathrm{d}\mathbb{P}_0}\frac{\mathrm{d}\mathbb{P}_0}{\mathrm{d}\mathbb{P}_{v_\theta}}(\boldsymbol{X}) \tag{15}$$

$$= \log 1_{\mathcal{B}}(\boldsymbol{X}) - \log Z - \int_0^T (\boldsymbol{v}_\theta^T \Sigma^{-1})(\boldsymbol{X}_t) \cdot \mathrm{d}\boldsymbol{X}_t \tag{16}$$

$$+ \int_0^T (\Sigma^{-1}\boldsymbol{u} \cdot \boldsymbol{v}_\theta)(\boldsymbol{X}_t)\mathrm{d}t + \frac{1}{2}\int_0^T \|\boldsymbol{v}_\theta(\boldsymbol{X}_t)\|^2\mathrm{d}t \tag{17}$$

$$= \log 1_{\mathcal{B}}(\boldsymbol{X}) - \log Z - \int_0^T (\boldsymbol{v}_\theta \cdot \tilde{\boldsymbol{v}})(\boldsymbol{X}_t)\mathrm{d}t \tag{18}$$

$$- \int_0^T \boldsymbol{v}_\theta(\boldsymbol{X}_t) \cdot \mathrm{d}\boldsymbol{W}_t + \frac{1}{2}\int_0^T \|\boldsymbol{v}_\theta(\boldsymbol{X}_t)\|^2\mathrm{d}t \tag{19}$$

$$= F_{v_\theta,\tilde{v}}(\boldsymbol{X}) - \log Z \tag{20}$$

Since $\log Z$ is the constant, it is canceled out in the log-variance divergence as follows:

$$\mathbb{E}_{\mathbb{P}_{\tilde{v}}}\left[\left(\log\frac{\mathrm{d}\mathbb{Q}}{\mathrm{d}\mathbb{P}_{v_\theta}} - \mathbb{E}_{\mathbb{P}_{\tilde{v}}}\left[\log\frac{\mathrm{d}\mathbb{Q}}{\mathrm{d}\mathbb{P}_{v_\theta}}\right]\right)^2\right] = \mathbb{E}_{\mathbb{P}_{\tilde{v}}}\left[(F_{v_\theta,\tilde{v}} - \mathbb{E}_{\mathbb{P}_{\tilde{v}}}[F_{v_\theta,\tilde{v}}])^2\right], \tag{21}$$

## A.2  CONNECTION TO EXISTING LOSS FUNCTIONS ON DISCRETE-TIME DOMAIN

In this section, we connect our discretized loss of Equation (8) to the loss function, called relative trajectory balance (Venkatraman et al., 2024, RTB). Like our methods, RTB also amortized inference in target path distribution by training forward distribution on discrete-time domains such as vision, language, and control tasks. When discretized, our loss function is equivalent to the RTB objective.

Our goal is to show that for every paths $\boldsymbol{x}_{0:L}$ sampled from the path measure $\mathbb{P}_{\boldsymbol{v}_{\bar{\theta}}}$,

$$(\hat{F}_{v_\theta,v_{\bar{\theta}}}(\boldsymbol{x}_{0:L}) - w)^2 = \left(\log\frac{p_0(\boldsymbol{x}_{0:L})1_{\mathcal{B}}(\boldsymbol{x}_{0:L})}{Z_\theta p_{v_\theta}(\boldsymbol{x}_{0:L})}\right)^2, \tag{22}$$

where $w = \log Z_\theta$ is a learnable scalar parameter, and path distribution $p_{\boldsymbol{v}}(\boldsymbol{x}_{0:L}) = \prod_{\ell=0}^{L-1} p_{\boldsymbol{v}}(\boldsymbol{x}_{l+1}|\boldsymbol{x}_l)$ is Markovian, and its transition kernel $p_{\boldsymbol{v}}(\boldsymbol{x}_{l+1}|\boldsymbol{x}_l)$ are derived from Euler-Maruyama discretization of the SDE in Equation (3) as follows:

$$\boldsymbol{x}_{l+1} = \boldsymbol{x}_l + \boldsymbol{u}(\boldsymbol{x}_l)\Delta t + \Sigma\boldsymbol{v}(\boldsymbol{x}_l)\Delta t + \Sigma\boldsymbol{\epsilon}_l, \tag{23}$$

where $\epsilon_l \sim \mathcal{N}(\mathbf{0}, \Delta t)$. To this end, we can calculate as follows:

$$\log p_0(\boldsymbol{x}_{0:L}) - \log p_{\boldsymbol{v}_\theta}(\boldsymbol{x}_{0:L}) \tag{24}$$

$$= \sum_{\ell=0}^{L-1} \log p_{\boldsymbol{v}_\theta}(\boldsymbol{x}_{l+1}|\boldsymbol{x}_l) - \sum_{\ell=0}^{L-1} \log p_0(\boldsymbol{x}_{l+1}|\boldsymbol{x}_l) \tag{25}$$

$$= \frac{1}{2}\sum_{\ell=0}^{L-1}(\Sigma\boldsymbol{v}_{\bar{\theta}}\Delta t + \Sigma\boldsymbol{\epsilon}_l - \Sigma\boldsymbol{v}_\theta\Delta t)^T(\Sigma^T\Sigma\Delta t)^{-1}(\Sigma\boldsymbol{v}_{\bar{\theta}}\Delta t + \Sigma\boldsymbol{\epsilon}_l - \Sigma\boldsymbol{v}_\theta\Delta t) \tag{26}$$

$$- \frac{1}{2}\sum_{\ell=0}^{L-1}(\Sigma\boldsymbol{v}_{\bar{\theta}}\Delta t + \Sigma\boldsymbol{\epsilon}_l)^T(\Sigma^T\Sigma\Delta t)^{-1}(\Sigma\boldsymbol{v}_{\bar{\theta}}\Delta t + \Sigma\boldsymbol{\epsilon}_l) \tag{27}$$

$$= \frac{1}{2\Delta t}\sum_{\ell=0}^{L-1}(\|\boldsymbol{v}_{\bar{\theta}}\Delta t + \boldsymbol{\epsilon}_l - \boldsymbol{v}_\theta\Delta t\|^2 - \|\boldsymbol{v}_{\bar{\theta}}\Delta t + \boldsymbol{\epsilon}_l\|^2) \tag{28}$$

$$= \frac{1}{2}\sum_{\ell=0}^{L-1}\|\boldsymbol{v}_\theta(\boldsymbol{x}_\ell)\|^2\Delta t - \sum_{\ell=0}^{L-1}(\boldsymbol{v}_\theta \cdot \boldsymbol{v}_{\bar{\theta}})(\boldsymbol{x}_\ell)\Delta t - \sum_{\ell=0}^{L-1}\boldsymbol{v}_\theta(\boldsymbol{x}_\ell) \cdot \boldsymbol{\epsilon}_\ell \tag{29}$$

$$= \hat{F}_{\boldsymbol{v}_\theta, \boldsymbol{v}_{\bar{\theta}}}(\boldsymbol{x}_{0:L}) - \log 1_{\mathcal{B}}(\boldsymbol{x}_{0:L}) \tag{30}$$

which implies

$$\hat{F}_{\boldsymbol{v}_\theta, \boldsymbol{v}_{\bar{\theta}}}(\boldsymbol{x}_{0:L}) = \log \frac{p_0(\boldsymbol{x}_{0:L})1_{\mathcal{B}}(\boldsymbol{x}_{0:L})}{p_{\boldsymbol{v}_\theta}(\boldsymbol{x}_{0:L})}, \tag{31}$$

by subtracting $w$ and squaring both sides, we have

$$(\hat{F}_{\boldsymbol{v}_\theta, \boldsymbol{v}_{\bar{\theta}}}(\boldsymbol{x}_{0:L}) - w)^2 = \left(\log \frac{p_0(\boldsymbol{x}_{0:L})1_{\mathcal{B}}(\boldsymbol{x}_{0:L})}{Z_\theta p_{\boldsymbol{v}_\theta}(\boldsymbol{x}_{0:L})}\right)^2 \tag{32}$$

We can view $p_0(\boldsymbol{x}_{0:L})1_{\mathcal{B}}(\boldsymbol{x}_{0:L})$ as the unnormalized target distribution discretized from the target path measure $\mathbb{Q}$, and $Z_\theta$ as the estimator for normalizing constant $Z = \int p_0(\boldsymbol{x}_{0:L})1_{\mathcal{B}}(\boldsymbol{x}_{0:L})d\boldsymbol{x}_{0:L}$, and $p_{\boldsymbol{v}_\theta}(\boldsymbol{x}_{0:L})$ as forward probability distribution to amortize inference in the target distribution. Based on these results, we provide our training algorithm in Algorithm 1.

### A.3 PROOF OF SCALE-BASED PARAMETERIZATION

In this section, we prove that our scale-based parameterization of bias force strictly decreases the distance to the (aligned) target position for small step sizes, improving the ability to find informative paths in large molecules.

**Proposition 1.** *Consider the molecular state $\boldsymbol{R}_t$ at the $t$-th time step and the next state $\boldsymbol{R}'_{t+\Delta t} = \boldsymbol{R}_t + \boldsymbol{b}(\boldsymbol{X}_t)\Delta t/\boldsymbol{m}$ updated by step size $\Delta t$ and the bias force $\boldsymbol{b}(\boldsymbol{X}_t) = diag(\boldsymbol{s}_\theta(\rho_t^{-1} \cdot \boldsymbol{X}_t))(\rho_t \cdot \boldsymbol{R}_{\mathcal{B}} - \boldsymbol{R}_t)$. Then there always exists a small enough $\Delta t$ that strictly decreases the distance towards the target state $\boldsymbol{R}_{\mathcal{B}}$:*

$$\|\rho'_{t+\Delta t} \cdot \boldsymbol{R}_{\mathcal{B}} - \boldsymbol{R}'_{t+\Delta t}\| < \|\rho_t \cdot \boldsymbol{R}_{\mathcal{B}} - \boldsymbol{R}_t\|, \tag{33}$$

*where $\rho'_{t+\Delta t} = \operatorname{argmin}_{\rho \in SE(3)}\|\rho \cdot \boldsymbol{R}_{\mathcal{B}} - \boldsymbol{R}'_{t+\Delta t}\|$ and we assume that there does not exist a rotation that exactly aligns the current molecular state to the target state, i.e., $\|\rho_t \cdot \boldsymbol{R}_{\mathcal{B}} - \boldsymbol{R}_t\| > 0$.*

*Proof.* The proof consists of two steps. We first show the (strictly) positive correlation between the bias force and the direction from the $t$-th state $\boldsymbol{R}_t$ to the target state $\boldsymbol{R}_{\mathcal{B}}$. Next, we show that the positive correlation gaurantees a strict decrease in distance between the states, i.e., $\|\rho_t \cdot \boldsymbol{R}_{\mathcal{B}} - \boldsymbol{R}_t\|$, given that the distance was not already zero.

**Step 1:** First, we show that the bias force (divided by atom-wise masses) is positively correlated with the direction to the target position, i.e., $(\boldsymbol{b}(\boldsymbol{X}_t)/\boldsymbol{m})^\top(\rho_t \cdot \boldsymbol{R}_{\mathcal{B}} - \boldsymbol{R}_t) > 0$. This follows from:

$$(\boldsymbol{b}(\boldsymbol{X}_t)/\boldsymbol{m})^\top(\rho_t \cdot \boldsymbol{R}_{\mathcal{B}} - \boldsymbol{R}_t) = (\rho_t \cdot \boldsymbol{R}_{\mathcal{B}} - \boldsymbol{R}_t)^T\frac{\operatorname{diag}(\boldsymbol{s}_\theta(\rho_t^{-1} \cdot \boldsymbol{X}_t))}{\boldsymbol{m}}(\rho_t \cdot \boldsymbol{R}_{\mathcal{B}} - \boldsymbol{R}_t) \tag{34}$$

$$= \sum_{i=1}^{3N}\left(\frac{\boldsymbol{s}_i}{\boldsymbol{m}_i}\right)(\rho_t \cdot \boldsymbol{R}_{\mathcal{B}} - \boldsymbol{R}_t)_i^2 > 0, \tag{35}$$

where $s_i > 0$ is the $i$-th element of $s_\theta(\rho_t^{-1} \cdot X_t)$ and $(\rho_t \cdot R_\mathcal{B} - R_t)_i$ is the $i$-th element of the direction to the target position.

**Step 2:** Next, we show that the positive correlation ensures distance reduction for a small enough step size. Consider the squared distance between the target position $\rho_t \cdot R_\mathcal{B}$ and updated position $R'$ by bias force

$$\|\rho_t \cdot R_\mathcal{B} - R'_{t+\Delta t}\|^2$$

$$= \|\rho_t \cdot R_\mathcal{B} - (R_t + b(X_t)\Delta t/m)\|^2 \tag{36}$$

$$= \|(\rho_t \cdot R_\mathcal{B} - R_t) - b(X_t)\Delta t/m\|^2 \tag{37}$$

$$= \|\rho_t \cdot R_\mathcal{B} - R_t\|^2 - 2\Delta t(b(X_t)/m)^\top (\rho_t \cdot R_\mathcal{B} - R_t) + (\Delta t)^2\|b(X_t)/m\|^2. \tag{38}$$

Due to step 1, i.e., $(b(X_t)/m)^\top (\rho_t \cdot R_\mathcal{B} - R_t) > 0$, there exists a step size $\Delta t$ satisfying:

$$0 < \Delta t < \frac{2(b(X_t)/m)^\top (\rho_t \cdot R_\mathcal{B} - R_t)}{\|b(X_t)/m\|^2}. \tag{39}$$

With this choice of $\Delta t$, multiplying $\Delta t\|b(X_t)/m\|^2$ leads to the following inequaliity:

$$(\Delta t)^2\|b(X_t)/m\|^2 < 2\Delta t(b(X_t)/m)^\top (\rho_t \cdot R_\mathcal{B} - R_t). \tag{40}$$

By subtracting the right-hand side from both sides and adding $\|\rho_t \cdot R_\mathcal{B} - R_t\|^2$ to both sides, we have the following inequality:

$$\|\rho_t \cdot R_\mathcal{B} - R'_{t+\Delta t}\|^2 < \|\rho_t \cdot R_\mathcal{B} - R_t\|^2. \tag{41}$$

Taking the square root of both sides, we have the following inequality:

$$\|\rho'_{t+\Delta t} \cdot R_\mathcal{B} - R'_{t+\Delta t}\| \le \|\rho_t \cdot R_\mathcal{B} - R'_{t+\Delta t}\| < \|\rho_t \cdot R_\mathcal{B} - R_t\|, \tag{42}$$

where the first inequality follows from the definition of $\rho'_{t+\Delta t} = \mathrm{argmin}_{\rho \in SE(3)}\|\rho \cdot R_\mathcal{B} - R'_{t+\Delta t}\|$. This completes the proof. $\square$

## B    EXPERIMENT DETAILS

### B.1    OPENMM CONFIGURATIONS

For real-world molecules, we use the VVVR integrator (Sivak et al., 2014) with the step size $\Delta t = 1$ fs and the friction term $\gamma = 1\,\mathrm{ps}^{-1}$. In the TPS-DPS training algorithm, we simulate MD with $T = 1\mathrm{ps}$ for Alanine Dipeptide and $T = 5\mathrm{ps}$ for the fast-folding proteins. We start simulations at a temperature $\lambda_{\mathrm{start}} = 600\mathrm{K}$, and end at a temperature $\lambda_{\mathrm{end}} = 300\mathrm{K}$ for Alanine Dipeptide and Chignolin and $\lambda_{\mathrm{end}} = 400\mathrm{K}$ for the others. We use the `amber99sbildn` (Lindorff-Larsen et al., 2010) force field for Alanine Dipeptide in vacuum and the `ff14SBonlysc` (Maier et al., 2015) force field for the fast-folding proteins with the `gbn2` implicit solvation model (Nguyen et al., 2013).

### B.2    MODEL CONFIGURATIONS

We use a 3-layer MLP for the double-well system, and a 6-layer MLP for real-world molecules with ReLU activation functions for neural bias force, potential, and scale. To constrain the output of the neural bias scale parameterization to a positive value, we apply Softplus to the MLP output. As an input to the neural network, we concatenate the current position $(\boldsymbol{R}_t)_i$ of the $i$-th atom with its distance to the target position $d_i = \|(\tilde{\boldsymbol{R}}_{\mathcal{B}})_i - (\boldsymbol{R}_t)_i\|_2$. For real-world molecules, we apply the Kabsch algorithm (Kabsch, 1976) for heavy atoms to align $\boldsymbol{R}_{\mathcal{B}}$ with $\boldsymbol{R}_t$. We update the parameters of the neural network with a learning rate of 0.0001, while the scalar parameter $w$ is updated with a learning rate of 0.001. We clip the gradient norm with 1 to prevent loss from exploding. we train $J = 1000$ times per rollout. We report other model configurations in Table 3. For PIPS, we use the model configurations reported by Holdijk et al. (2024). For CVs of SMD, we use backbone dihedral angles $(\phi, \psi)$ for Alanine Dipeptide and RMSD for fast-folding proteins.

Table 3: **Model configurations of TPS-DPS.**

| System | # of rollouts ($I$) | # of samples ($M$) | Batch size ($K$) | Buffer size | Relaxation ($\sigma$) |
|---|---|---|---|---|---|
| Double-well | 20 | 512 | 512 | 10000 | 3 |
| Alanine Dipeptide | 1000 | 16 | 16 | 1000 | 0.1 |
| Chignolin | 100 | 16 | 4 | 200 | 0.5 |
| Trpcage | 100 | 16 | 4 | 100 | 0.5 |
| BBA | 100 | 16 | 4 | 100 | 0.5 |
| BBL | 100 | 16 | 2 | 100 | 0.5 |

### B.3    EVALUATION METRICS

**Root mean square distance (RMSD).** We use the Kabsch algorithm (Kabsch, 1976) for heavy atoms to align the final position with the target position $\boldsymbol{R}_{\mathcal{B}}$, using the optimal (proper) rotation and translation to superimpose two heavy atom positions. We calculate RMSD between heavy atoms of the final position and the target position $\boldsymbol{R}_{\mathcal{B}}$.

**Target hit percentage (THP).** THP measures the success rate of paths arriving at the target meta-stable state $\mathcal{B}$ in a binary manner. Formally, given the final positions $\{R^{(i)}\}_{i=1}^{M}$ of $M$ paths, THP is defined as follows:

$$\mathrm{THP} = \frac{|\{i : \boldsymbol{R}^{(i)} \in \mathcal{B}\}|}{M} \tag{43}$$

**Energy of transition state (ETS).** ETS measures the ability of the method to find probable transition states when crossing the energy barrier. ETS refers to the maximum potential energy among states in a transition path. Formally, given a transition path $\boldsymbol{x}_{0:L}$ of length $L$ that reaches the target meta-stable state i.e., $\boldsymbol{R}_L \in \mathcal{B}$, ETS is defined as follows:

$$\mathrm{ETS}(\boldsymbol{x}_{0:L}) = \max_{\ell \in [0,L]} U(\boldsymbol{R}_\ell) \tag{44}$$

### B.4 SYSTEM DETAILS

**Systhetic double-well system.** Double-well system follows the overdamped Langevin dynamics defined as follows:

$$d\boldsymbol{R}_t = \frac{-\nabla U(\boldsymbol{R}_t)}{\boldsymbol{m}}dt + \sqrt{\frac{2\gamma k_B \lambda}{\boldsymbol{m}}}d\boldsymbol{W}_t \,. \tag{45}$$

For simplicity, we let $\boldsymbol{R} = (x, y) \in \mathbb{R}^2$, $\boldsymbol{m} = I, \gamma = 1, \Delta = 0.01, T = 10$, and $\lambda = 1200\text{K}$. To evaluate the ability to find diverse transition paths, we consider the following double-well potential (Hua et al., 2024):

$$U(x, y) = \frac{1}{6}\left(4(1 - x^2 - y^2)^2 + 2(x^2 - 2)^2 + [(x + y)^2 - 1]^2 + [(x - y)^2 - 1]^2 - 2\right) \,. \tag{46}$$

This potential has global minima and two saddle points, having two meta-stable states and two reaction channels.

**Time-lagged independent components (TICA).** To extract the collective variable (CV) for the four fast-folding proteins, we consider components of time-lagged independent component analysis (TICA; Pérez-Hernández et al., 2013). We run $1\mu$s unbiased MD simulations with 2fs step size and record states per 2ps to collect MD trajectories, using the OpenMM library with the same configuration as in Appendix B.1. For the top two TICA components, we use PyEMMA library (Scherer et al., 2015) with a time lag $\tau = 500\text{ps}$ for Chignolin $\tau = 200\text{ps}$ for the others.

**Reproducibility.** We describe experiment details in Appendix B, including detailed simulation configuration and hyper-parameters. In the anonymous link, we provide the code for TPS-DPS.

## C COMPUTATIONAL COST

In this section, we analyze the time complexity of TPS-DPS and provide the number of energy evaluations and runtime in training and inference time for real molecules.

The training and inference time complexity of TPS-DPS is $O(NMLJ)$ and $O(NML)$, respectively, where $N$ is the number of atoms, $M$ is the number of samples, $L$ is the number of MD steps, and $J$ is the number of rollouts. To be specific, training consists of biased MD simulations with $O(NML)$ time complexity. Given $M$, the total complexity of one biased MD step is $O(N)$.

To justify it, we note that the biased MD step consists of three stages: (1) calculating bias force, (2) calculating OpenMM force field, and (3) integrating the biased MD. Given the number of layers, and hidden units, MLP for bias force requires $O(N)$ and the Kabsch algorithm for equivariance requires $O(N)$. Calculating force with cut-off and integrating MD with VVVR integrator also requires $O(N)$.

To measure computational cost, we consider the number of energy evaluations and runtime per rollout in training and inference time. As shown in Table 4, the inference cost of TPS-DPS is proportional to UMD and SMD which have time complexity $O(NML)$. TPS-DPS requires less energy evaluations than PIPS in training since TPS-DPS finds transition paths faster than PIPS by utilizing the replay buffer and simulated annealing.

Table 4: **Cost comparison.** EET and EEI refer to the number of energy evaluations by the OpenMM library in training and inference, respectively. RT and RI denote runtime (second) per rollout in training and inference on a single RTX A5000 GPU. MD simulations are conducted with $T = 1$ps for Alanine Dipeptide, $T = 5$ps for other systems, and $\Delta t = 1$fs. The number of samples for each training rollout and inference is 16 and 64, respectively. Other configurations of TPS-DPS are provided in Appendix B.2 and those of PIPS can be found in (Holdijk et al., 2024)

| Molecule | Method | EET ($\downarrow$) | EEI ($\downarrow$) | RT ($\downarrow$) | RI ($\downarrow$) |
|---|---|---|---|---|---|
| Alanine Dipeptide | UMD | - | 64K | - | 29.49 |
| | SMD | - | 64K | - | 47.45 |
| | PIPS (F) | 240M | 64K | 44.22 | 71.05 |
| | PIPS (P) | 240M | 64K | 50.54 | 75.67 |
| | TPS-DPS (F, Ours) | 16M | 64K | 24.93 | 70.50 |
| | TPS-DPS (P, Ours) | 16M | 64K | 27.25 | 78.83 |
| | TPS-DPS (S, Ours) | 16M | 64K | 25.11 | 73.04 |
| Chignolin | UMD | - | 320K | - | 224.23 |
| | SMD | - | 320K | - | 380.37 |
| | PIPS (F) | 40M | 320K | 277.29 | 565.58 |
| | PIPS (P) | 40M | 320K | 317.08 | 622.87 |
| | TPS-DPS (F, Ours) | 8M | 320K | 209.29 | 562.90 |
| | TPS-DPS (P, Ours) | 8M | 320K | 224.36 | 623.63 |
| | TPS-DPS (S, Ours) | 8M | 320K | 215.18 | 581.26 |
| Trp-Cage | UMD | - | 320K | - | 258.29 |
| | SMD | - | 320K | - | 323.52 |
| | PIPS (F) | 40M | 320K | 360.55 | 652.59 |
| | PIPS (P) | 40M | 320K | 417.93 | 718.05 |
| | TPS-DPS (F, Ours) | 8M | 320K | 289.10 | 655.22 |
| | TPS-DPS (P, Ours) | 8M | 320K | 301.76 | 699.44 |
| | TPS-DPS (S, Ours) | 8M | 320K | 293.51 | 673.00 |
| BBA | UMD | - | 320K | - | 395.12 |
| | SMD | - | 320K | - | 551.04 |
| | PIPS (F) | 40M | 320K | 506.57 | 1080.62 |
| | PIPS (P) | 40M | 320K | 574.07 | 1117.74 |
| | TPS-DPS (F, Ours) | 8M | 320K | 422.23 | 1042.81 |
| | TPS-DPS (P, Ours) | 8M | 320K | 430.24 | 1091.97 |
| | TPS-DPS (S, Ours) | 8M | 320K | 426.48 | 1068.68 |
| BBL | UMD | - | 320K | - | 673.55 |
| | SMD | - | 320K | - | 853.77 |
| | PIPS (F) | 40M | 320K | 622.22 | 1558.53 |
| | PIPS (P) | 40M | 320K | 660.02 | 1612.78 |
| | TPS-DPS (F, Ours) | 8M | 320K | 560.95 | 1520.05 |
| | TPS-DPS (P, Ours) | 8M | 320K | 572.77 | 1607.62 |
| | TPS-DPS (S, Ours) | 8M | 320K | 563.45 | 1553.89 |

# D VISUALIZATION OF SAMPLED PATHS OF FAST FOLDING PROTEINS

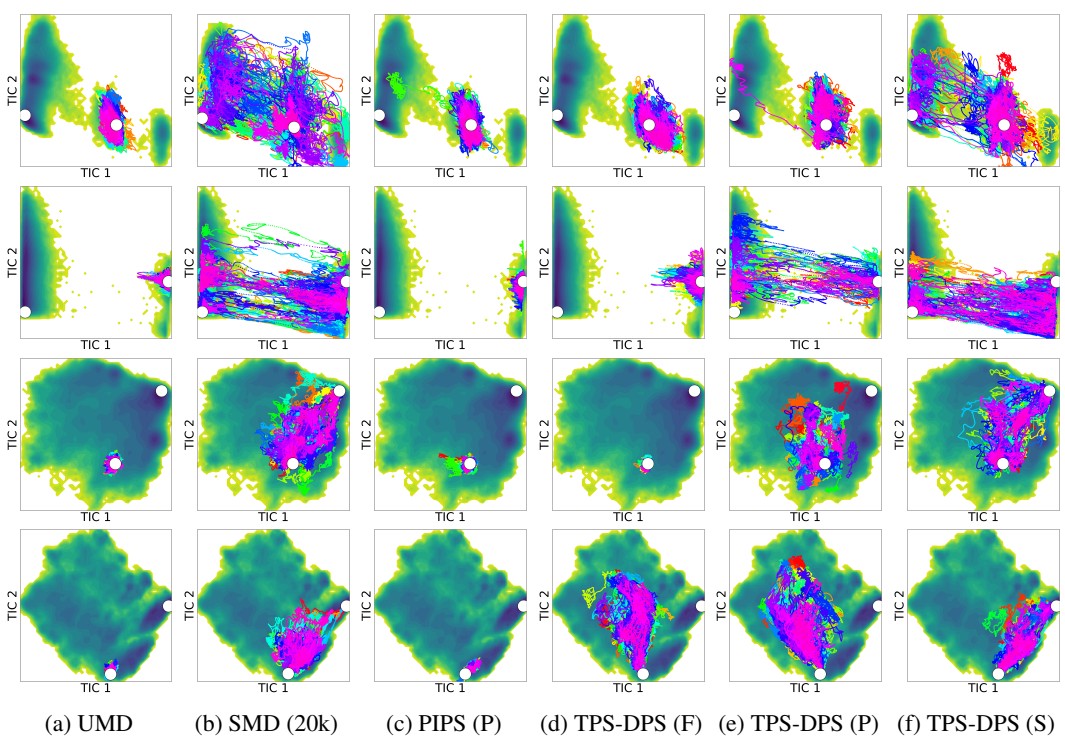

(a) UMD  (b) SMD (20k)  (c) PIPS (P)  (d) TPS-DPS (F)  (e) TPS-DPS (P)  (f) TPS-DPS (S)

Figure 7: **Visualization of sampled paths of four fast-folding proteins on free energy landscapes for top two TICA components.** We aim to sample folding processes for the four fast folding proteins: Chignolin, Trpcage, BBA, and BBL (from top to bottom rows).

# E    COMPARISION WITH REVERSE KL DIVERGENCE

Table 5: **Benchmark scores of reverse KL divergence and TPS-DPS on Alanine Dipeptide system**. Metrics are averaged over 64 paths. TPS-DPS consistently outperforms reverse KL divergence on all metrics regardless of predicting bias force or potential.

| Method | RMSD ($\downarrow$) Å | THP ($\uparrow$) % | ETS ($\downarrow$) kJmol$^{-1}$ |
|---|---|---|---|
| Reverse KL (F) | $0.43 \pm 0.34$ | 53.12 | $27.88 \pm 14.38$ |
| Reverse KL (P) | $0.58 \pm 0.34$ | 48.43 | $21.61 \pm 11.76$ |
| TPS-DPS (F, Ours) | $\mathbf{0.16 \pm 0.06}$ | **92.19** | $19.82 \pm 15.88$ |
| TPS-DPS (P, Ours) | $\mathbf{0.16 \pm 0.10}$ | 87.50 | $\mathbf{18.37 \pm 10.86}$ |

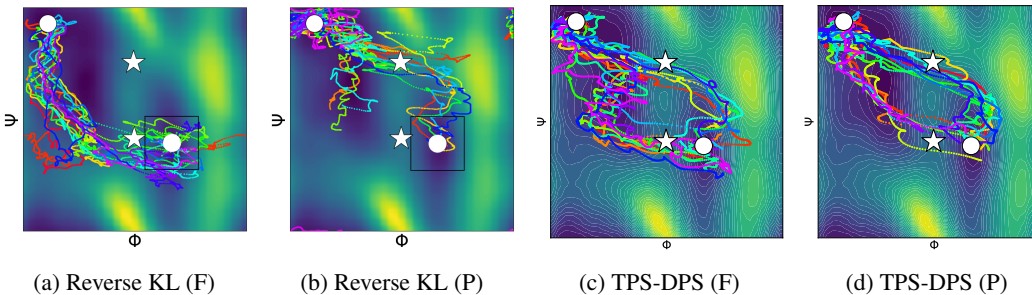

(a) Reverse KL (F)          (b) Reverse KL (P)          (c) TPS-DPS (F)          (d) TPS-DPS (P)

Figure 8: **Visualization of sampled paths of Alanine Dipeptide on the Ramachandran plot.** The reverse KL divergence struggles to find diverse reaction channels suffering from mode collapse issues while the log-variance divergence of our method can capture two reaction channels and reach the target states better.

