# OpenReview forum: "Transition Path Sampling with Improved Off-Policy Training of Diffusion Path Samplers"
_ICLR.cc/2025/Conference — ICLR 2025 Poster_

### Official Review · Reviewer_aeFr · 2024-10-20

**Soundness:** 3
**Presentation:** 2
**Contribution:** 2
**Rating:** 6
**Confidence:** 3

**Summary:**

The authors present Transition Path Sampling with Diffusion Path Sampling (TPS-DPS) an approach to accelerate the sampling of transition pathways in molecular systems. As is somewhat standard, this is formulated as learning a potential which biases the original (Langevin) dynamics in order to facilitate the transition. The authors formulate the problem as minimizing a log-variance divergence between the target path measure (i.e transition path ensemble of unbiased MD simulations) and the path measure induced by the biased dynamics. This is claimed to be better than a KL divergence objective in terms of variance reduction. The core contribution of the paper is to formulate an off-policy, RL-style training workflow in which trajectories are sampled from the biased dynamics and added to a growing replay buffer, and  the log variance divergence is continuously minimized along with a learnable control variate. A special bias parameterization is chosen in order to promote more frequent transitions. Results are demonstrated on several model systems, including a double well potential, alanine dipeptide, the polyproline helix, and chignolin folding. The authors demonstrate that they outperform non-ML and ML baselines in terms of the diversity and quality of the sampled transition paths.

**Strengths:**

1. The authors are tackling an important and challenging problem that has received less attention from the ML community compared to topics like machine learning force fields, etc, although that is now changing.
2. The core method is fairly simple and straightforward to implement, and the results on the model systems are strong relative to the baselines.

**Weaknesses:**

My primary concerns center around the lack of clarity in presentation, and not considering higher-dimensional systems (other than a brief mention at the end of the paper). I will elaborate further.

Higher-Dimensional Systems:

I will preface this critique with the following disclaimer: I am aware that the systems considered in this work (especially alanine dipeptide and chignolin) are widely used as models to study transition states (including in the PIPS paper which the authors directly compare to), and I understand why the authors chose to demonstrate their methods on these systems. However, given the growing set of ML methods in this area which all benchmark on these same tasks, standout methods are those which demonstrate potential to generalize to more realistic, higher-dimensional systems. At the end of this paper, the authors acknowledge that “we did not evaluate our algorithm for even larger proteins due to the lack of computational budget. In addition, our work does not generalize across unseen pairs of meta-stable states or different molecular systems”. To me, this is not sufficient.

The authors claim that they don’t scale up due to a lack of computational budget. However, it’s hard to place this statement in context with the rest of the paper, since as far as I could tell, no training cost/runtime/FLOP analysis was provided for the existing systems. How does the training cost scale with the dimensionality of the system and/or the height of energy barriers between metastable states? Is it truly just a scale problem, or will the method fundamentally break down for larger systems due to the difficulty in sampling informative trajectories for off-policy learning? This could also be explored theoretically: how does the variance of the reweighting procedure grow as system dimensionality grows and transition states become harder to find? I would like to see an investigation of these questions on a higher-dimensional system (perhaps one of the larger fast folding proteins from Lindorf-Larssen et. al. [1]). Even if the method breaks down, a failure mode analysis would be a valuable addition to the literature on ML for transition path sampling, which has so far been characterized by many papers “declaring victory” on simple systems. This would help the community more realistically assess the promise of ML-based TPS. If this is done convincingly and well, I would be willing to reconsider my evaluation.

Lack of Clarity of Presentation

Several aspects of the paper were confusing to follow.
1. I did not understand why the decomposition of the bias force into a component aligned with the direction of the target state and a component orthogonal to it is useful (Section 3.3). Do you only retain the component which is in the direction of the target state? Further clarification on the motivation and benefits of this decomposition would be helpful. I believe removing this decomposition is tested in the Ablations, but only in terms of the final loss, not in terms of the effect it has on sampling informative trajectories during training. A figure illustrating the effect of this decomposition could make this a lot more clear.

2. The KL divergence loss is mentioned several times as an alternative to the proposed log variance divergence objective, but is never explicitly written. As I understand it, the only difference between the loss function (Eqn 10) and a KL objective is that the expectation can be evaluated w.r.t an old policy (i.e off-policy learning). I think it would help to directly contrast the two objectives, perhaps with a side-by-side comparison.

3. Quoting from the paper, "The first three terms in Equation (6) correspond to the deviation of the biased MD from the unbiased
MD integrated over the path sampled from Pv ". I interpret the first term as a regularization of the bias force to be small, which makes sense. However, the second and third terms are not clear to me. In the second term, should this be interpreted as a dot product between the forces from the old and new policy? And the interpretation for integrating the third term against the Brownian motion is not clear to me. As the quantity in Equation 6 appears many times throughout the paper, I recommend the authors more carefully break down and explain each of the components more clearly.

4. Personally, I think the logical flow of the paper would be enhanced if the training algorithm was placed in the main text, as it is concisely encapsulates the training procedure.

A Few Other Miscellaneous Points:

1. When evaluating the quality of transition paths, it would be useful to compare to the ground truth transition paths obtained from e.g. long, unbiased MD simulations. For example, for chignolin, these are obtainable from reference simulations performed by D.E. Shaw [1].

2. A citation of Sipka, et. al. [2] would be well-placed as they also tackle TPS with ML.

**Questions:**

1. Why are MLPs chosen to parameterize the bias forces for the molecular systems? Have you tried using simple GNNs?

2. Is it possible to use the replay buffer to serve as positive examples for “virtual” goal states (e.g. Hindsight Experience Replay [3])? I.e, even if a path does not reach the target state, it does reach some other state, and it can be treated as a positive/expert demonstration for reaching that state.

3. As far as I could tell, ground truth MD data is not used anywhere in the method. I understand that unbiased MD rarely achieves transitions, but in the situations that it does, could you leverage these “expert demonstrations” to improve your method?

[1] Lindorff-Larsen, K., Piana, S., Dror, R. O., & Shaw, D. E. (2011). How fast-folding proteins fold. Science, 334(6055), 517-520.

[2] Sipka, M., Dietschreit, J. C., Grajciar, L., & Gómez-Bombarelli, R. (2023, July). Differentiable simulations for enhanced sampling of rare events. In International Conference on Machine Learning (pp. 31990-32007). PMLR.

[3] Andrychowicz, M., Wolski, F., Ray, A., Schneider, J., Fong, R., Welinder, P., ... & Zaremba, W. (2017). Hindsight experience replay. Advances in neural information processing systems, 30.

---

> ### Author Response · Authors · 2024-11-23
>
> Dear reviewer aeFr,
>
> We express our deep appreciation for your time and insightful comments. In our updated manuscript, we highlight the changes in Blue.
>
> In what follows, we address your comments one by one.
> ___
>
> **W1: The authors mention lack of computational budget, but the paper lacks analysis of training cost or scaling behaviour. Investigating scalability of the method through theoretical or empirical investigation on higher-dimensional systems would be valuable.**
>
> We sincerely appreciate this feedback. We mentioned the lack of a computational budget due to our personal constraint at the time of submission, however, we fully agree with your comment that the thorough investigation would be valuable.
>
> To this end, we investigate the scaling behavior of our method to larger proteins, analyze the computational complexity of our method, and report the training and inference cost. We conclude by pointing out the limitations of our method for much larger real-world problems, which is added in the limitation section of our updated manuscript.
>
> **Scaling behavior of our method to larger proteins.** We investigate scaling behavior and observed that our method succeeds even in proteins such as Trpcage, BBA, and BBL from [1] that are larger than the previously considered Chignolin by two to five times. However, even with crude approximation, our method is not applicable to medium proteins (approximately at least 1700 GPU hours for medium proteins) or explicit solvent settings without investing a significant amount of computation.
>
> We provide the quantitative results of scaling behavior of our method to Trpcage, BBA, and BBL in Table A and Appendix D of our updated manuscript. We also provide a video of the generated trajectory on the [project page](https://anonymous.4open.science/w/tps-dps-0941/).
>
> ###### Table A. Benchmark scores on Trpcage, BBA and BBL at 400K. All metrics are averaged over 64 paths.
> | Trpcage                  | RMSD ↓ | THP ↑ | ETS ↓|
> |-------------------------|-------|-------|------|
> | UMD | 7.94 | 0.00 | - |
> | UMD (1200K)| 8.27 | 0.00 | - |
> | SMD (10k)| 1.68 | 3.12 | -312.54 |
> | SMD (20k)| 1.20 | 41.29 | -226.40 |
> | TPS-DPS (F, Ours)   | 6.35  | 0.00  | - |
> | TPS-DPS \(P, Ours)   | 3.15  | 12.50 | **-512.97** |
> | TPS-DPS (S, Ours)   | **0.76**  | **81.25**  | -317.61 |
>
> | BBA                 | RMSD ↓ | THP ↑ | ETS ↓|
> |-------------------------|-------|-------|------|
> | UMD | 10.03 | 0.00 | - |
> | UMD (1200K)| 10.81 | 0.00 | - |
> | SMD (10k)| 2.89 | 0.00 | - |
> | SMD (20k)| 1.66 | 26.56 | -3104.95 |
> | TPS-DPS (F, Ours)   | 9.48  | 0.00  | - |
> | TPS-DPS \(P, Ours)   | 3.89  | 0.00 | - |
> | TPS-DPS (S, Ours)   | **1.21**  | **84.38**  | **-3801.68** |
>
> | BBL                 | RMSD ↓ | THP ↑ | ETS ↓|
> |-------------------------|-------|-------|------|
> | UMD | 18.48 | 0.00 | - |
> | UMD (1200K)| 18.90 | 0.00 | - |
> | SMD (10k)| 3.67 | 0.00 | -
> | SMD (20k)| 2.97 | 7.81 | -1738.57 |
> | TPS-DPS (F, Ours)   | 10.15 | 0.00 | - |
> | TPS-DPS \(P, Ours)  | 6.45 | 0.00 | - |
> | TPS-DPS (S, Ours)   | **1.60** | **43.75** | **-3616.32** |
>
> Surprisingly, one can observe that our method with scale-based parameterization still outperforms the considered baselines even for larger proteins. Even when inspecting the trajectories provided in our [project page](https://anonymous.4open.science/w/tps-dps-0941/), we find the motions to be quite realistic. However, we are not declaring victory, as we point out later.

---

> ### Author Response · Authors · 2024-11-23
>
> **Time complexity of training.** The training and inference time complexity of our approach is $O(NMLJ)$ and $O(NML)$, respectively. ($N$ is the number of atoms, $M$ is the number of samples, $L$ is the number of MD steps, and $J$ is the number of rollouts.) To be specific, training consists of biased MD simulations with $O(NML)$ time complexity. Given the number of samples $M$, the time complexity of one biased MD step of TPS-DPS is $O(N)$.
>
> To justify it, we note that the biased MD step consists of three stages: (1) calculating bias force, (2) calculating OpenMM force field, and (3) integrating the biased MD. Given the number of layers, and hidden units, MLP for bias force requires $O(N)$ and the Kabsch algorithm for equivariance requires $O(N)$. Calculating the force field with cut-off and integrating MD with VVVR integrator also requires $O(N)$.
>
> **Computational cost of experiments in the paper.** To measure computational cost, we consider the number of energy evaluations which is proportional to the number of rollouts and runtime per rollout in training and inference time. As shown in the following Table A, TPS-DPS requires less energy evaluations than PIPS [2] in training time, producing informative paths within several rollouts. The inference cost of our approach is proportional to Steered MD (SMD) while ML methods require training for higher performance than SMD.
>
> ###### Table A. Comparision of computational costs with PIPS and SMD in training and inference time. EET and EEI denote the number of energy evaluations in training and inference time, respectively. RT and RI denote runtime per rollout in training and inference time, respectively.
> | Molecule|method                  | EET ↓ | RT ↓ | EEI ↓| RI ↓|
> |------|-------------------|-------|-------|-------|-------
> |       |           |  | s | | s|
> |ALDP |SMD |-|-|64K|47.45|
> |ALDP |PIPS (F) | 240M | 44.22 | 64K | 71.05 |
> |ALDP |TPS-DPS (F)   | 16M  | 24.93  | 64K | 70.50|
> |Chignolin|SMD | - | - | 320K | 283.45 |
> |Chignolin| PIPS (F) | 40M | 553.82 | 320K | 565.58
> |Chignolin |TPS-DPS (F)   | 8M | 209.29 | 320K | 562.90|
> |Trpcage |SMD | - | - | 320K | 323.52 |
> |Trpcage |TPS-DPS (F)   | 8M | 289.10 | 320K | 655.22 |
> |BBA |SMD | - | - | 320K | 542.35 |
> |BBA |TPS-DPS (F)   | 8M | 422.23 | 320K | 1042.81 |
> |BBL |SMD | - | - | 320K | 853.77 |
> |BBL |TPS-DPS (F)   | 8M | 560.95 | 320K | 1520.05 |
>
> TPS-DPS is a computationally efficient alternative to PIPS, with significantly reduced training costs while maintaining inference costs comparable to SMD. This efficiency makes it suitable for generating paths across both small and large molecular systems.
>
>
> **Limitation of our method.** While we have provided some promising results, we emphasize that we are not declaring any victory since the experimented proteins (<50 amino acids) are still small. Our main bottleneck is the cost of running multiple MD simulations. On a single A5000 GPU, training our method for BBL with 47 amino acids requires around a week since it requires 560.95 (sec) per rollout as shown in Table A. Training our models for much larger proteins (> 500 amino acids) would require at least 10 weeks. Moreover, we do not consider explicit solvent since this would require much more computations for calculating the force fields than implicit solvent.

---

> ### Author Response · Authors · 2024-11-23
>
> **W1-1. This could also be explored theoretically: How does the variance of the reweighting procedure grow as system dimensionality grows?**
>
> We would like to point out that the log-variance divergence doesn't require importance weight due to the freedom in the reference path measure. Thus, the variance of the reweighting procedure of our method is 0 regardless of system dimensionality.
> ___
> **W2-1. Further clarification on the motivation and benefits of the decomposition (scale-based parameterization) would be helpful. A figure illustrating the effect of this decomposition (scale-based parameterization) on sampling informative trajectories would be helpful.**
>
> First, we clarify that the bias force decomposition illustrates the effect of the positive correlation of bias force from scale-based parameterization with the direction to the target position. There is no explicit decomposition of bias force in our algorithm. To avoid confusion, we removed the explanation about force decomposition in our updated manuscript.
>
> The motivation for scale-based parameterization is the failure of bias force and bias potential parameterizations in sampling informative paths during training. The key benefit of this parameterization is the guarantee of reducing distance to the target position for every MD step as proved in Appendix A.3 of our updated manuscript. This allows bias force to sample transition paths during training in larger systems.
>
> To resolve your concern, we illustrate the effect of scale-based parameterization in Figure 2 of our updated manuscript. The figure describes bias forces positively correlated with the target directions in positive scaling parameterization.
> ___
>
> **W2-2. The KL divergence loss is mentioned several times as an alternative to the proposed log variance divergence objective, but is never explicitly written. As I understand it, the only difference between the loss function (eq 10) and a KL objective is that the expectation can be evaluated w.r.t an old policy (i.e off-policy learning). I think it would help to directly contrast the two objectives, perhaps with a side-by-side comparison.**
>
> To resolve your concern, we directly compare two objectives side-by-side in the following Table C and Appendix E. The discretized form of KL divergence (or relative entropy) and proposed log-variance divergence (Eqn 10) are explicitly written as follows:
> $$
>     \mathcal{L}_\text{KL}(\theta)=\mathbb{E} _{x _{0:L} \sim p _{v _{\theta}}(x _{0:L})}\left[\log\frac{p _{0}(x _{0:L})1 _{\mathcal{B}}(x _{0:L})}{p _{v _{\theta}}(x _{0:L})}\right],
> $$
>
> $$\mathcal{L} _\text{LV}(\theta, w)=\mathbb{E} _{x _{0:L} \sim \hat{\mathcal{D}}}\left[\left(\log\frac{p _{0}(x _{0:L})1 _{\mathcal{B}}(x _{0:L})}{p _{v _{\theta}}(x _{0:L})} - w\right)^{2}\right].
> $$
>
> As shown in the following Table C, log-variance divergence (1) reuses samples from old policies with replay buffer, (2) uses an annealing technique to explore and reach the target well, (3) detaches gradients from the SDE solver, and (4) reduces the variance of gradient estimator with learnable control variate $w$. These benefits are impossible for on-policy loss, e.g. reverse KL divergence.
>
> ###### Table C. Side-by-side comparison of reverse KL and log-variance divergence (eq 10).
> |                  | KL | Log-variance |
> |-------------------------|-------|-------|
> | Reusing samples from old policy               | X  | O |
> | Exploring and target reaching with annealing | X | O |
> | Avoiding computational graphs of SDE solver | X | O |
> | Reducing gradient variance with control variate $w$| X | O |
> |
>
> In Appendix E of our updated manuscript, we also provide the quantitative and qualitative results to demonstrate that the proposed loss leads to more accurate and diverse transition paths than reverse KL divergence.

---

> ### Author Response · Authors · 2024-11-23
>
> **W2-3. I recommend the authors more carefully break down and explain each of the components in eq (6) more clearly.**
>
> To resolve your concern, we provide two different interpretations.
>
> Eq (6) in our submitted manuscript is as follows:
> \begin{align}
> F _{v _{\theta}, \tilde{v}}(X)&= \frac{1}{2}\int ^{T} _{0}\lVert v _{\theta}(X _{t})\rVert ^{2}\mathrm{d}t-\int ^{T} _{0}(v _{\theta} \cdot \tilde{v})(X _{t})\mathrm{d}t\\\\
> &\quad -\int ^{T} _{0} v _{\theta}(X _{t}) \cdot \mathrm{d}W _{t}+ \log 1 _{\mathcal{B}}(X)
> \end{align}
>
> The first interpretation is based on Radon-Nikodym derivatives. By applying the Girsanov theorem, eq (6) can be rewritten as
> \begin{align}
> F _{v _{\theta}, \tilde{v}}(X)&= \frac{1}{2}\int^{T} _{0}\lVert v _{\theta}(X _{t})\rVert^{2}\mathrm{d}t-\int^{T} _{0}(v _{\theta}\cdot \tilde{v})(X _{t})\mathrm{d}t \\\\
>     &\quad -\int^{T} _{0}v _{\theta}(X _{t}) \cdot \mathrm{d}W _{t}+\log 1 _{\mathcal{B}}(X)\\\\
>     &= {\log\frac{\mathrm{d}\mathbb{P} _{0}}{\mathrm{d}\mathbb{P} _{v _{\theta}}}(X)}+{\log\frac{\mathrm{d}\tilde{\mathbb{Q}}}{\mathrm{d}\mathbb{P} _{0}}(X)},
> \end{align}
>
> here, $\tilde{\mathbb{Q}}$ is the (unnormalized) target path measure such that $\frac{\mathrm{d}\tilde{\mathbb{Q}}}{\mathrm{d}\mathbb{P} _{0}}(X)=1 _{\mathcal{B}}(X)$. The first three terms of RHS in eq (6) correspond to ${\log\frac{\mathrm{d}\mathbb{P} _{0}}{\mathrm{d}\mathbb{P} _{v _{\theta}}}(X)}$, the (log-likelihood) deviation of the path measure induced by biased MD from that by unbiased MD. The last term $1 _{\mathcal{B}}(X)$ corresponds to $\frac{\mathrm{d}\tilde{\mathbb{Q}}}{\mathrm{d}\mathbb{P} _{0}}(X)$, connecting path measure induced by unbiased MD and the (unnormalized) target path measure by reweighting the path measure depending on the success in reaching target.
>
> The second interpretation is based on policies. The first term $\frac{1}{2}\int^{T} _{0}\lVert v _{\theta}(X _{t})\rVert^{2}\mathrm{d}t$ can be interpreted as regularization term for bias force. The second and third terms, rewritten as follows
> $$
> \int^{T} _{0}v _{\theta}(X _{t})\cdot\left(\tilde{v}(X _{t}) + \frac{\mathrm{d}W _{t}}{\mathrm{d}t}\right)\mathrm{d}t=\left<v _\theta,\tilde{v}+\frac{\mathrm{d}W _{t}}{\mathrm{d}t}\right>,
> $$
> can be interpreted as the similarity between current policy $v _\theta$ and noised policy of $\tilde{v}$ for the reference path measure.
> ___
>
> **W2-4. Personally, I think the logical flow of the paper would be enhanced if the training algorithm was placed in the main text, as it concisely encapsulates the training procedure.**
>
> We agree with this suggestion and moved the training algorithm to the main text in our updated manuscript.
>
> ___
> **W3. When evaluating the quality of transition paths, it would be useful to compare to the ground truth transition paths obtained from e.g. long, unbiased MD simulations. For example, for chignolin, these are obtainable from reference simulations performed by D.E. Shaw.**
>
> We tried this, but concluded that the trajectories are incomparable due to different simulation settings. D.E. Shaw [1] deployed MD trajectories with intervals of 200ps, which is too coarse compared to our step size of 1fs. Moreover, we experiment on implicit solvent settings, unlike D.E.Shaw.
> ___
> **W4. A citation of Sipka, et. al. would be well-placed as they also tackle TPS with ML.**
>
> To resolve your concern, in our updated manuscript, we place Sipka et al., [3] in related works. They used differentiable biased MD simulation to train bias potential and introduced partial back-propagation and graph mini-batching techniques to resolve computational issues in differentiable simulation.

---

> ### Author Response · Authors · 2024-11-23
>
> **Q1. Why are MLPs chosen to parameterize the bias forces for the molecular systems? Have you tried using simple GNNs?**
>
> We used MLP for the fair comparison to PIPS [2] which also used MLP. We tried to use EGNN [4] before and found that MLP seems to outperform EGNN in small systems. For larger systems, applying GNN seems to be more promising.
>
> **Q2 Is it possible to use the replay buffer to serve as positive examples for “virtual” goal states (e.g. Hindsight Experience Replay)? I.e, even if a path does not reach the target state, it does reach some other state, and it can be treated as a positive/expert demonstration for reaching that state.**
>
> Yes, Hindsight experience replay (HER) [5] can be used to augment target states. We tried it before and found that it doesn't seem to work well given only two meta-stable states of a molecule. If you are interested in training a single model for various meta-stable states and molecular systems, HER will be an effective design choice for the replay buffer.
>
> **Q3. As far as I could tell, ground truth MD data is not used anywhere in the method. I understand that unbiased MD rarely achieves transitions, but in the situations that it does, could you leverage these “expert demonstrations” to improve your method?**
>
> Yes, we tried to pre-train the model with small amounts of expert trajectories and then apply our approaches. While it seems to accelerate convergence, it does not improve final performances in the synthetic system.
>
>
> **References**
>
> [1] Lindorff-Larsen, Kresten, et al. "How fast-folding proteins fold." Science 334.6055 (2011): 517-520.
>
> [2] Holdijk, Lars, et al. "Stochastic optimal control for collective variable free sampling of molecular transition paths." Advances in Neural Information Processing Systems 36 (2024).
>
> [3] Sipka, Martin, et al. "Differentiable simulations for enhanced sampling of rare events." International Conference on Machine Learning. PMLR, 2023.
>
> [4] Satorras, Vıctor Garcia, Emiel Hoogeboom, and Max Welling. "E (n) equivariant graph neural networks." International conference on machine learning. PMLR, 2021.
>
> [5] Andrychowicz, Marcin, et al. "Hindsight experience replay." Advances in neural information processing systems 30 (2017).
>
> ___

---

> > ### Comment · Reviewer_aeFr · 2024-11-24
> > **Response**
> >
> > Thanks for the detailed responses. The authors have mostly addressed my concerns regarding the lack of results on larger systems, as well as a more detailed runtime/scaling analysis, by including experiments on Trp-cage, BBA, and BBL from the D.E. Shaw fast folding proteins dataset. It appears that the transition states found by TPS-DPS in this setting are fairly reasonable.
> >
> > A couple of minor comments:
> >
> > 1. The authors mention they tried pre-training with small amounts of expert trajectories and then applying their approach, but it did not improve the final performance. I would like to see this mentioned in the paper, perhaps with a Figure in the Appendix.
> > 2. Can the authors comment on whether coarse-graining could make the problem setting more scalable, and whether there are any issues applying their method when considering coarse-grained protein models?
> >
> > In light of the changes, I have raised my score to a 6. The reason I am not raising it higher is that the very long training/inference times for small/moderately sized proteins, plus a lack of obvious ways to use this method to generalize across many diverse proteins/atomistic systems, makes the method feel somewhat impractical for wide adoption at the moment (for instance, if you ran unbiased simulations of BBL at 400K on a reasonably parallelized machine for an entire week - which is the training time the authors report - would you not see some transition events?) However, I appreciate the changes and think the paper would be interesting to the audience at ICLR.

---

> > > ### Author Response · Authors · 2024-11-29
> > >
> > > Dear Reviewer aeFr,
> > >
> > > Thank you for the response! We think your comments were very helpful in improving our paper. We appreciate your insightful comments to improve our paper.
> > >
> > > In what follows, we address your additional comments one by one.
> > > ___
> > >
> > > **M1. The authors mention they tried pre-training with small amounts of expert trajectories and then applying their approach, but it did not improve the final performance. I would like to see this mentioned in the paper, perhaps with a Figure in the Appendix.**
> > >
> > > We would like to clarify that our observation about pre-training applies specifically to the double-well synthetic system, where ground-truth data collection is possible. In this case, pre-training with small amounts of expert trajectories does not significantly improve the final performance.
> > >
> > > To address your request, we have included the loss curves over rollouts and quantitative/qualitative results for pre-training with 1024 ground-truth transition paths in Appendix F. As shown in Table D and Figure 13, in the double-well system, pre-training with 1024 ground-truth transition paths leads to lower initial loss but does not significantly improve the final performance.
> > >
> > > ###### Table D. Ablation studies on pre-training with 1024 ground-truth transition paths in the double-well synthetic system at 1200K.
> > > | Method                 | RMSD ↓ | THP ↑ | ETS ↓|
> > > |-------------------------|-------|-------|------|
> > > | TPS-DPS (F)             | 0.01  | 99.90  | 1.38 |
> > > | TPS-DPS (F) w/ pre-train  | 0.01  | 100.00  | 1.37 |
> > > ___
> > >
> > > **M2. Can the authors comment on whether coarse-graining could make the problem setting more scalable, and whether there are any issues applying their method when considering coarse-grained protein models?**
> > >
> > > Coarse-graining can indeed enhance the scalability of our approach. By defining the biasing forces on coarse-grained beads, our method is naturally compatible with coarse-grained molecular dynamics. This allows for efficient training and inference on larger molecular systems that would be computationally prohibitive in an all-atom simulation framework.
> > >
> > > However, the success of applying our method to coarse-grained models depends critically on the quality of the coarse-grained representation. It is essential that the coarse-grained model captures the relevant slow degrees of freedom and that the coarse-grained potential is consistent with the underlying free energy surface. Inaccuracies in the coarse-grained representation could affect the reliability of the results.
> > >
> > > We view this as an exciting direction for future research.

---

> > > ### Author Response · Authors · 2024-11-29
> > >
> > > **W1-1. The very long training/inference times for small/moderately sized proteins make the method feel somewhat impractical for wide adoption at the moment (for instance, if you ran unbiased simulations of BBL at 400K on a reasonably parallelized machine for an entire week - which is the training time the authors report - would you not see some transition events?)**
> > >
> > > Thank you for highlighting the practical considerations of training and inference times in the context of broader adoption.
> > >
> > > First, we would like to clarify that the inference time for our method is fast and comparable to unbiased MD simulations, as demonstrated in our time complexity analysis. We agree that the long training time is our current limitation. However, we would like to point out that the "long" training time is still much faster than unbiased MD as shown in Tables D and E. Furthermore, once we train our model, we can easily generate many transition paths while unbiased MD still requires crossing the energy barrier.
> > >
> > > ###### Table D. Comparision with variable length unbiased MD in GPU hours to obtain 1000 transition paths of Chignolin (300K) and BBL (400K).
> > > | Chignolin (300K)                 | GPU hours |
> > > |------------------------|-----------|
> > > | TPS-DPS (S)            | 60 (training) +3 (inference)|
> > > | variable length unbiased MD            | 648      |
> > > |
> > >
> > > | BBL (400K)                 | GPU hours |
> > > |------------------------|-----------|
> > > | TPS-DPS (S)            | 156 (training) +7 (inference)|
> > > | variable length unbiased MD            | 6000      |
> > > |
> > >
> > > ###### Table E. The number of transition paths of Chignolin (300K) and BBL (400K) which TPS-DPS (S) and variable length unbiased MD can sample during a week on a single A5000 GPU.
> > > | Chignolin (300K)                  | The number of transition paths |
> > > |------------------------|-----------|
> > > | TPS-DPS (S)            | 66607     |
> > > | variable length unbiased MD            | 37        |
> > > |
> > >
> > > | BBL (400K)               | The number of transition paths |
> > > |------------------------  |-----------|
> > > | TPS-DPS (S)              | 1791     |
> > > | variable length unbiased MD |  4     |
> > > |
> > >
> > >
> > > ___
> > >
> > > **W1-2 a lack of obvious ways to use this method to generalize across many diverse proteins/atomistic systems**
> > >
> > > We believe our method can be generalized to other systems by replacing the MLP-based architecture with a Geometric GNN architecture. One can simply train our algorithm across multiple proteins, similar to training a Boltzmann generator that generalizes across proteins [1].
> > > ___
> > >
> > > **References**
> > >
> > > [1] Klein, Leon, and Frank Noé. "Transferable Boltzmann Generators." arXiv preprint arXiv:2406.14426 (2024).

---

### Official Review · Reviewer_ALy8 · 2024-11-03

**Soundness:** 3
**Presentation:** 3
**Contribution:** 3
**Rating:** 6
**Confidence:** 3

**Summary:**

This paper introduces an approach that trains diffusion path samplers for transition path sampling without requiring collective variables. To ensure scalability in high-dimensional tasks, the authors propose an off-policy training objective that incorporates learning control variates with replay buffers and a scale-based equivariant parameterization of bias forces. The proposed TPS-DPS method has been evaluated on a synthetic double-well potential and three peptides.

**Strengths:**

The paper is clearly written and contributes to the important drug discovery task of understanding transition pathways between meta-stable states in molecular systems. The design of CV-free diffusion path sampler using log-variance divergence with the learnable control variate is novel to me. For the experimental part, the visualizations of the sampling paths effectively demonstrate the accuracy and diversity of the sampled transition paths. The ablation study thoroughly illustrates the effectiveness of each design component.

**Weaknesses:**

My main concern is with the baselines included in the experiments. Presently, only one ML baseline, PIPS, is included. The authors have mentioned several other baselines in the related work section, such as Jing et al. (2024), Das et al. (2021), Hua et al. (2024), and Du et al. (2024). Including these in the experiments would be beneficial.

**Questions:**

In the introduction section, a reference is needed for the claim that "minimizing the KL divergence leads to mode collapse, ...".

---

> ### Author Response · Authors · 2024-11-23
>
> Dear reviewer ALy8,
>
> We express our deep appreciation for your time and insightful comments. In our updated manuscript, we highlight the changes in Blue.
>
> In what follows, we address your comments one by one.
>
> ---
>
> **W1. My main concern is with the baselines included in the experiments. Presently, only one ML baseline, PIPS, is included. The authors have mentioned several other baselines in the related work section, such as Jing et al. (2024), Das et al. (2021), Hua et al. (2024), and Du et al. (2024). Including these in the experiments would be beneficial.**
>
> To resolve your main concern, in our updated manuscript, we compare with two-way shooting (non-ML baseline) and Doob's Lagrangian [1] (ML baseline) in Alanine Dipeptide at 300K. As shown in the following Table A, our approach produces more plausible transition states than baselines and finds the transition states of various potential energies, unlike Doob's Lagrangian.
>
> ###### Table A. Comparision with variable length two-way shooting and Doob's Lagrangian with internal coordinate and 2 mixtures in Alanine Dipeptide at 300K.
> | method                  | ETS ↓|
> |-------------------------|------|
> | Two-way shooting        | 527.66 $\pm$ 450.51 |
> | Doob's Lagrangian       | 69.26 $\pm$ 0.21 |
> | TPS-DPS (F, Ours)       | 19.82 $\pm$ 15.88|
> | TPS-DPS \(P, Ours)      | **18.37 $\pm$ 10.86** |
>
> Other baselines are not comparable to our method. Jing et al [2]., generated time-coarsened transition paths with step size 10ps which is too coarsened to capture transition states while we use small steps size 1fs. They also required datasets generated by long unbiased MD simulation to train the model while we can train the model without datasets collected in advance. Das et al [3]., and Hua et al [4]., limited their evaluation to low-dimensional synthetic systems and did not consider real molecular systems.
> ___
> **Q1. In the introduction section, a reference is needed for the claim that "minimizing the KL divergence leads to mode collapse, ...".**
>
> To resolve your concern, in the introduction section of our updated manuscript, we provide the reference [5, 6] to claim that reverse KL divergence suffers from mode collapse.
> ___
>
> **References**
>
> [1] Du, Yuanqi, et al. "Doob's Lagrangian: A Sample-Efficient Variational Approach to Transition Path Sampling." The Thirty-eighth Annual Conference on Neural Information Processing Systems.
>
> [2] Jing, Bowen, et al. "Generative Modeling of Molecular Dynamics Trajectories." The Thirty-eighth Annual Conference on Neural Information Processing Systems.
>
> [3] Das, Avishek, et al. "Reinforcement learning of rare diffusive dynamics." The Journal of Chemical Physics 155.13 (2021).
>
> [4] Hua, Xinru, et al. "Accelerated Sampling of Rare Events using a Neural Network Bias Potential." arXiv preprint arXiv:2401.06936 (2024).
>
> [5] Vargas, Francisco, Will Sussman Grathwohl, and Arnaud Doucet. "Denoising Diffusion Samplers." The Eleventh International Conference on Learning Representations.
>
> [6] Richter, Lorenz, and Julius Berner. "Improved sampling via learned diffusions." arXiv preprint arXiv:2307.01198 (2023).

---

> > ### Comment · Reviewer_ALy8 · 2024-11-27
> > **Response**
> >
> > Thanks for the detailed response. I don't have further questions.

---

> > > ### Author Response · Authors · 2024-11-29
> > >
> > > Thank you for your response. We think your comments were very helpful in improving our paper.

---

### Official Review · Reviewer_r5xj · 2024-11-04

**Soundness:** 3
**Presentation:** 3
**Contribution:** 3
**Rating:** 8
**Confidence:** 3

**Summary:**

The paper proposes a set of improvements to existing data-free machine learning methods for sampling transition paths. They show that avoiding KL divergence loss, using a replay buffer, and other changes all produce positive results over previous methods on small molecular test systems.

**Strengths:**

The lop-variance minimization is a clear improvement over KL-based minimization. Similarly, the off-policy training scheme with a replay buffer also improves things. For evaluating these contributions, figure 9 is clear and strong.

The results on all of the toy systems evaluated are strong.

**Weaknesses:**

While the paper provides clear improvements over previous methods, it is not clear that the improvements enable the method to tackle the grand challenge of finding plausible transition paths for real biomolecular systems. Beyond this, there is not much negative to say about the method. The results for chignolin in figure 6 are a bit lacking. It is difficult to draw conclusions from only 16 paths, but I certainly understand the computational limitations here.

The paper could be clearer in laying out the overall problem. Without reading the previous papers in this area, it was difficult to follow. (eg. common terms like "path measure" could be more clearly defined)

**Questions:**

What is the computational cost compared to SMD?
What limitations prevent you from applying this method to larger systems?

---

> ### Author Response · Authors · 2024-11-23
>
> Dear Reviewer r5xj,
>
> We express our deep appreciation for your time and insightful comments. In our updated manuscript, we highlight the changes in Blue.
>
> In what follows, we address your comments one by one.
>
> ___
>
> **W1. While the paper provides clear improvements over previous methods, it is not clear that the improvements enable the method to tackle the grand challenge of finding plausible transition paths for real biomolecular systems.**
>
> We agree that, at the moment, our work does not solve the grand challenge of finding plausible transition paths for real biomolecular systems, such as medium-sized proteins in explicit solvent settings with all-atom simulations. This would require significant algorithmic developments and inverestment of computational resources that are currently unavailable in most academic setting.
>
> Nevertheless, to further emphasize the promise of our method on scaling to real-world biomolecular systems, we experiment on fast proteins of larger size in Appendix C of our updated manuscript. We experiment on Trpcage, BBA, and BBL with 20, 28, and 47 amino acids, respectively, which are significantly larger than the previously considered Chignolin with 10 amino acids. As shown in the following Table A, our approach with scale-based parameterization outperforms the considered baselines.
>
> ###### Table A. Benchmark scores on Trpcage, BBA and BBL at 400K. All metrics are averaged over 64 paths.
> | Trpcage                  | RMSD ↓ | THP ↑ | ETS ↓|
> |-------------------------|-------|-------|------|
> | UMD | 7.94 | 0.00 | - |
> | UMD (1200K)| 8.27 | 0.00 | - |
> | SMD (10k)| 1.68 | 3.12 | -312.54 |
> | SMD (20k)| 1.20 | 41.29 | -226.40 |
> | TPS-DPS (F, Ours)   | 6.35  | 0.00  | - |
> | TPS-DPS \(P, Ours)   | 3.15  | 12.50 | **-512.97** |
> | TPS-DPS (S, Ours)   | **0.76**  | **81.25**  | -317.61 |
>
> | BBA                 | RMSD ↓ | THP ↑ | ETS ↓|
> |-------------------------|-------|-------|------|
> | UMD | 10.03 | 0.00 | - |
> | UMD (1200K)| 10.81 | 0.00 | - |
> | SMD (10k)| 2.89 | 0.00 | - |
> | SMD (20k)| 1.66 | 26.56 | -3104.95 |
> | TPS-DPS (F, Ours)   | 9.48  | 0.00  | - |
> | TPS-DPS \(P, Ours)   | 3.89  | 0.00 | - |
> | TPS-DPS (S, Ours)   | **1.21**  | **84.38**  | **-3801.68** |
>
> | BBL                 | RMSD ↓ | THP ↑ | ETS ↓|
> |-------------------------|-------|-------|------|
> | UMD | 18.48 | 0.00 | - |
> | UMD (1200K)| 18.90 | 0.00 | - |
> | SMD (10k)| 3.67 | 0.00 | -
> | SMD (20k)| 2.97 | 7.81 | -1738.57 |
> | TPS-DPS (F, Ours)   | 10.15 | 0.00 | - |
> | TPS-DPS \(P, Ours)  | 6.45 | 0.00 | - |
> | TPS-DPS (S, Ours)   | **1.60** | **43.75** | **-3616.32** |
>
> We also plot 64 sampled paths on the top two TICA components with free energy surface in Appendix C of our updated manuscript and provide 3D videos of the sampled transition paths in our updated [project page](https://anonymous.4open.science/w/tps-dps-0941/). For experimental settings, refer to Appendix B.1 of our updated manuscript.
> ___
> **W2. It is difficult to draw conclusions from only 16 paths, but I certainly understand the computational limitations here.**
>
> This was not due to computational limitations but for better visualization. To resolve your concern, in Appendix C of our updated manuscript, we plot 64 sampled paths of fast-folding proteins on the top two TICA components with free energy surfaces.
>
> ___
> **W3. The paper could be clearer in laying out the overall problem. Without reading the previous papers in this area, it was difficult to follow. (eg. common terms like "path measure" could be more clearly defined)**
>
> To clarify the mathematical term, in the method section of our updated manuscript, we provide additional descriptions for path measure and Radon-Nikodym derivative.

---

> ### Author Response · Authors · 2024-11-23
>
> **Q1. What is the computational cost compared to SMD?**
>
> **Time complexity.** The training time complexity of our approach is $O(NMLJ)$ while the time complexity of Steered MD (SMD) is $O(NML)$. ($N$ is the number of atoms, $M$ is the number of samples, $L$ is the number of MD steps, and $J$ is the number of rollouts.) To be specific, training consists of biased MD simulations with $O(NML)$ time complexity. Given the number of samples $M$, the time complexity of one biased MD step of TPS-DPS is $O(N)$
>
> To justify it, we note that the biased MD step consists of three stages: (1) calculating bias force, (2) calculating OpenMM force field, and (3) integrating the biased MD. Given the number of layers, and hidden units, MLP for bias force requires $O(N)$ and the Kabsch algorithm for equivariance requires $O(N)$. Calculating the force field with cut-off and integrating MD with VVVR integrator also requires $O(N)$.
>
> **Empirical cost.** To measure computational cost, we consider the number of energy evaluations and runtime per rollout in training and inference time. As shown in the following Table B, the inference cost of our approach is proportional to SMD. The additional runtime in inference is from predicting bias forces and assigning them to external force objects of OpenMM Library. Note that our method requires training for higher performance than SMD.
>
> ###### Table B. Comparision of computational costs with SMD in training and inference time. EET and EEI denote the number of energy evaluation in training and inference time, respectively. RT and RI denote runtime per rollout in training and inference time, respectively.
> | Molecule|method                  | EET ↓ | RT ↓ | EEI ↓| RI ↓|
> |------|-------------------|-------|-------|-------|-------
> |       |           |  | s | | s|
> |ALDP |SMD |-|-|64K|47.45|
> |ALDP |TPS-DPS (F)   | 16M  | 24.93  | 64K | 70.50|
> |Chignolin|SMD | - | - | 320K | 283.45
> |Chignolin |TPS-DPS (F)   | 8M | 209.29 | 320K | 562.90|
> |Trpcage |SMD | - | - | 320K | 323.52 |
> |Trpcage |TPS-DPS (F)   | 8M | 289.10 | 320K | 655.22 |
> |BBA |SMD | - | - | 320K | 542.35 |
> |BBA |TPS-DPS (F)   | 8M | 422.23 | 320K | 1042.81 |
> |BBL |SMD | - | - | 320K | 853.77 |
> |BBL |TPS-DPS (F)   | 8M | 560.95 | 320K | 1520.05 |
>
> In Appendix D of our updated manuscript, we further compare the number of energy evaluations and runtime with the other ML baselines.
>
> ___
> **Q2. What limitations prevent you from applying this method to larger systems?**
>
> Our computational limitation is MD simulation with $O(NML)$ time complexity. Given the number of samples $M=16$, $L=5000$ steps, $J=1000$ rollouts, and a single A5000 GPU, training our method for BBL protein with 47 amino acids requires around a week since it requires 560.95 (sec) per rollout. Training our models for much larger proteins (> 500 amino acids) requires at least 10 weeks.

---

> > ### Comment · Reviewer_r5xj · 2024-11-25
> >
> > The authors have partially addressed my concerns about accuracy in scaling up to larger proteins. I have raised my score accordingly.

---

> > > ### Author Response · Authors · 2024-11-29
> > >
> > > Thank you very much for the positive response. We are happy to hear that we have addressed your concerns! We think your comments were very helpful in improving our paper.

---

### Official Review · Reviewer_CJA8 · 2024-11-04

**Soundness:** 3
**Presentation:** 3
**Contribution:** 2
**Rating:** 6
**Confidence:** 3

**Summary:**

This paper addresses the transition path sampling problem in molecular systems, a critical challenge in materials design and drug discovery. Traditional unbiased molecular dynamics (MD) simulations are computationally infeasible, while existing machine learning approaches often rely heavily on domain-specific knowledge. In response, the authors propose a diffusion path sampler that trains a neural network by minimizing the log-variance divergence between the path measure induced by biased MD and the target path measure.

**Strengths:**

1. The paper is well-written and easy to follow.

2. The use of log-variance as the training objective provides a more robust gradient estimator, eliminating the need to differentiate through the SDE solver. This approach also mitigates issues with the traditional KL-divergence objective, such as mode collapse.

3. A thorough empirical study demonstrates the effectiveness of the proposed method, and the comprehensive ablation study justifies each design choice.

4. Code is provided, enhancing the reproducibility of the results

**Weaknesses:**

1. The paper's novelty is somewhat limited, primarily focusing on the use of the log-variance objective. A deeper theoretical analysis would strengthen the contribution.

2. The term "diffusion path sampler" may imply a link to diffusion models, which could mislead readers. It would be helpful to clarify if there is a relationship between the proposed method and diffusion models, as the term "diffusion" only appears twice in the manuscript.

3. More discussion of concurrent work, particularly Doob’s Lagrangian: A Sample-Efficient Variational Approach to Transition Path Sampling, would provide valuable context.

**Questions:**

Please see the weakness section.

---

> ### Author Response · Authors · 2024-11-23
>
> Dear Reviewer CJA8,
>
> We express our deep appreciation for your time and insightful comments. In our updated manuscript, we highlight the changes in Blue.
>
> In what follows, we address your comments one by one.
>
> ---
> **W1. The paper's novelty is somewhat limited, primarily focusing on the use of the log-variance objective. A deeper theoretical analysis would strengthen the contribution.**
>
> We would like to gently point out that our novelty is not limited to simply using the existing log-variance divergence. We modify the original log-variance divergence [1] with learnable control variate and off-policy training to improve sample-efficiency and diversity. We also propose a novel scale-based parameterization of bias force which guarantees to decrease the distance to the target state given a small MD step size, sampling informative trajectories in large molecules.
>
> To further alleviate your concern about the lack of theoretical analysis, in Appendix A.3, we provide a theoretical analysis of the proposed scale-based parameterization. We show that (1) bias force with scale-based parameterization is positively correlated with the direction to the target position and (2) there exists a small MD step size such that the bias force reduces the distance to the target position.
>
> ___
>
> **W2. The term "diffusion path sampler" may imply a link to diffusion models, which could mislead readers. It would be helpful to clarify if there is a relationship between the proposed method and diffusion models, as the term "diffusion" only appears twice in the manuscript.**
>
> Thank you for pointing out the possible confusion. We clarify the term *diffusion path sampler* in the footnote of our updated introduction as follows:
>
> We coin our method *diffusion path sampler*, since it samples paths using diffusion SDE. Our notation closely aligns with *diffusion samplers* [2, 3] that samples from the final state (instead of the entire path) using diffusion SDE.
>
> ___
>
> **W3. More discussion of concurrent work, particularly Doob’s Lagrangian: A Sample-Efficient Variational Approach to Transition Path Sampling, would provide valuable context.**
>
> To incorporate your comment, in the related work section of our updated manuscript, we discuss more about the Doob’s Lagrangian [4] as follows:
>
> Doob's Lagrangian considered the TPS problem as finding Doob's $h$ transformation via the Lagrangian method with boundary constraints. They parameterized marginal distribution as (mixture) Gaussian path distribution to satisfy the boundary constraints without relying on simulation in training time and sampled transition paths with the bias force derived from the Fokker-Planck equation in inference time.
>
> In the updated experiments, we also compare with Doob's Lagrangian in Alanine Dipeptide at 300K. As shown in the following Table A, our approach produces more plausible transition states than baseline and finds the transition states of various potential energy.
>
> ###### Table A. Comparision with  Doob's Lagrangian with internal coordinate and 2 mixtures of Gaussian in Alanine Dipeptide at 300K.
> | method                  | ETS ↓|
> |-------------------------|------|
> | Doob's Lagrangian       | 69.26 $\pm$ 0.21 |
> | TPS-DPS (F, Ours)       | 19.82 $\pm$ 15.88|
> | TPS-DPS \(P, Ours)      | **18.37 $\pm$ 10.86** |
> ___
>
> **References**
>
> [1] Nüsken, Nikolas, and Lorenz Richter. "Solving high-dimensional Hamilton–Jacobi–Bellman PDEs using neural networks: perspectives from the theory of controlled diffusions and measures on path space." Partial differential equations and applications 2.4 (2021): 48.
>
> [2] Zhang, Qinsheng, and Yongxin Chen. "Path Integral Sampler: A Stochastic Control Approach For Sampling." International Conference on Learning Representations.
>
> [3] Vargas, Francisco, Will Sussman Grathwohl, and Arnaud Doucet. "Denoising Diffusion Samplers." The Eleventh International Conference on Learning Representations.
>
> [4] Du, Yuanqi, et al. "Doob's Lagrangian: A Sample-Efficient Variational Approach to Transition Path Sampling." The Thirty-eighth Annual Conference on Neural Information Processing Systems.

---

> > ### Comment · Reviewer_CJA8 · 2024-11-25
> >
> > Thank you for the detailed response. I don't have any further questions and am inclined to accept

---

> > > ### Author Response · Authors · 2024-11-29
> > >
> > > Thank you for your response. We think your comments were very helpful in improving our paper.

---

### Author Response · Authors · 2024-11-23

# General response

We sincerely thank all reviewers for their valuable and insightful comments. We highlight the updates of manuscript in blue. The key updates are as follows:

### Summary of key updates
- In Appendix C and our updated [project page](https://anonymous.4open.science/w/tps-dps-0941/), we provide quantitative and qualitative experiments on three fast folding proteins larger than Chingolin: Trpcage (Tryptophan-cage), BBA (Beta-beta-alpha) and BBL (a protein domain of biotin ligase enzyme). We also provide the model parameters in the updated [code](https://anonymous.4open.science/r/tps-dps-0941/).
- In Appendix D, we analyze the time complexity of our approach. We provide the number of energy evaluations and runtime spent in training and inference.
- In Appendix A.3, we prove that the bias force of positive scaling parameterization guarantees to decrease in the distance to the target state for a small MD step size.

---

### Meta-Review · Area_Chair_v3mV · 2024-12-20

**Metareview:**

The paper proposes a novel method for Transition Path Sampling, i.e. given the stochastic dynamic that defines the evolution of the system; the task is to sample the trajectories between endpoints (the initial and the final states of the system) that are distributed according to the corresponding measure. The main contribution of the paper is the optimization of the path measure (via the parameterized drift function) via the log-variance divergence, which (unlike the KL-divergence) allows for various Monte Carlo estimates. The proposed algorithm is supported by an empirical study.

**Additional Comments On Reviewer Discussion:**

Reviewers r5xj and aeFr raised concerns regarding the scalability of the approach, while Reviewers ALy8 and CJA8 proposed to compare with additional baselines. Most of the concerns were addressed by the authors during the rebuttal, which amounted to an overall positive evaluation of the paper.

---

### Decision · Program_Chairs · 2025-01-22

Accept (Poster)